# Over three decades, and counting, of near-surface turbulent flux measurements from the Atmospheric Radiation Measurement (ARM) user facility

Ryan C. Sullivan[1], David P. Billesbach[2,6], Sebastien Biraud[3], Stephen Chan[3], Richard Hart[1,6,⚱], Evan Keeler[1], Jenni Kyrouac[1], Sujan Pal[1], Mikhail Pekour[4], Sara L. Sullivan[5], Adam Theisen[1], Matt Tuftedal[1], and David R. Cook[1,6]

[1]Environmental Science Division, Argonne National Laboratory, Lemont, IL, 60439 USA
[2]Department of Biological Systems Engineering, School of Natural Resources, University of Nebraska, Lincoln, NE, 68583 USA
[3]Lawrence Berkeley National Laboratory, Berkeley, CA, 94720 USA
[4]Pacific Northwest National Laboratory, Richland, WA, 99352 USA
[5]independent researcher
[6]retired
⚱deceased

*Correspondence to:* Ryan Sullivan (rcsullivan@anl.gov)

**Abstract.** Processes mediating the coupling of terrestrial, aquatic, biospheric, and atmospheric systems influence weather, climate, and ecosystem dynamics via transfer of energy, momentum, water, and carbon (or other species). These exchange processes are quantified by measurements of near surface turbulent fluxes. Understanding processes at these interfaces provides insight into understanding and predicting current and future states within the Earth system. The Atmospheric Radiation Measurement (ARM) user facility has been conducting measurements of near surface turbulent fluxes since the early 1990's at long term fixed locations and shorter-term, mobile deployments across the Earth. ARM has utilized two established methods for conducting these measurements, energy balance Bowen ratio (EBBR) and eddy covariance (EC). Primary measurements from the former include sensible and latent heat flux, while the latter also measures fluxes of momentum and carbon (primarily carbon dioxide, with methane fluxes measured at two locations to date). The EBBR systems were deployed at 22 locations, and to date, the EC systems have been deployed at over 50 sites with plans for additional novel site locations into the future. Herein, the history, evolution, and key aspects of these instrument systems are documented, along with information on data quality assurance and post-processing, and best use practices. Additionally, three recent data validation experiments were conducted, and their key findings are summarized. Finally, ancillary datasets acquired by ARM, that can contextualize and aid interpretation of the near surface turbulent flux measurements, are discussed.

Datasets described herein include the eddy correlation flux measurement system: 30ECOR (https://doi.org/10.5439/1879993, Sullivan et al., 1997), 30QCECOR (https://doi.org/10.5439/1097546, Gaustad 2023), and ECORSF (https://doi.org/10.5439/1494128, Sullivan et al., 2019a); the energy balance Bowen ratio system: 30EBBR

(https://doi.org/10.5439/1023895, Sullivan et al., 1993) and 30BAEBBR (https://doi.org/10.5439/1027268, Gaustad and Xie
1993); and the carbon dioxide flux measurement system: CO2FLX (https://doi.org/10.5439/1287574,
https://doi.org/10.5439/1287575, https://doi.org/10.5439/1287576, Koontz et al., 2015a,b,c; https://doi.org/10.5439/1989774,
https://doi.org/10.5439/1989776, https://doi.org/10.5439/1992202, Biraud & Chan, 2002a,b,c). These data can be found by
searching the above datastream names at https://adc.arm.gov/discovery/#/results/.

## 1 Introduction

Knowledge of near surface turbulent fluxes (hereafter simply "fluxes"), the transport of quantities across the land(water)-
atmosphere-biosphere interface by turbulent eddies, are critical in understanding sources and sinks of energy, water, and
other atmospheric constituents (e.g., carbon, nitrogen, or sulfur species, and aerosol particles) (Yang et al., 2023). In addition
to modulating the aforementioned budgets, sensible and latent heat fluxes (H and LE, respectively; see "Appendix A
Acronyms and abbreviations") prescribe the evolution of the overlaying atmospheric boundary layer, impacting weather
locally and downwind (Helbig et al., 2021). While these fluxes can be estimated globally from satellite-based radiance
measurements coupled with theoretical models, in situ meteorology, or numerical Earth system models, these methods are
potentially subject to large uncertainties and often fail to capture the fine spatial scales at which these processes occur (Chu
et al., 2021; Ershadi et al., 2014; Sullivan et al., 2019b,c; Velpuri et al., 2013). Thus, in situ measurements of H and LE are
necessary to fill this knowledge gap, and their information content is critical for understanding and predicting processes
relevant for heatwaves, drought monitoring, wildfire response and prescribed burn planning, agriculture and irrigation
scheduling, freshwater management, and the anthropogenic drivers therein (Fisher et al., 2017; Miralles et al., 2019).
Furthermore, flux measurements of carbon, coupled with LE, are critical in understanding biologic system processes, their
controls and trends, and predicting changes in these processes in the future (Baldocchi et al., 2024).

Since its inception, the U.S. Department of Energy (DOE) Atmospheric Radiation Measurement (ARM) user facility (Stokes
& Schwartz, 1994; Turner & Ellingson, 2016) has measured fluxes primarily using an in situ meteorologically-driven,
energy balance flux gradient method with the Energy Balance Bowen Ratio (EBBR) system (Cook & Sullivan, 2025a), and
the eddy covariance method (EC) with the Carbon Dioxide Flux (CO2FLX) measurement system (Chan & Biraud, 2022)
and the Eddy Correlation (ECOR) flux measurement system (Cook & Sullivan, 2025b). With the mission to improve
understanding and modeling of atmospheric processes in global climate models (GCMs) and Earth system models (ESMs),
these systems have been deployed at various long- and short-term sites globally, including at more heavily instrumented,
spatially distributed sites across the central USA (Fig. 1; Table B1). These data have been used extensively to study a range
of topics within Earth system science including but not limited to: land-atmosphere interactions and impacts of land surface
heterogeneity on atmospheric processes (e.g., Feldman et al., 2023; Phillips et al., 2017; Tian et al., 2022), surface energy
budgets (e.g., Liu et al., 2025; Oehri et al., 2022), arctic carbon exchange (e.g., Bao et al., 2021; Zolkos et al., 2022),

boundary layer and convective processes (e.g., Daub & Lareau, 2022; Wakefield et al., 2023), and to validate and improve

earth          systems          models          (e.g.,          Qin          et          al.,          2023).

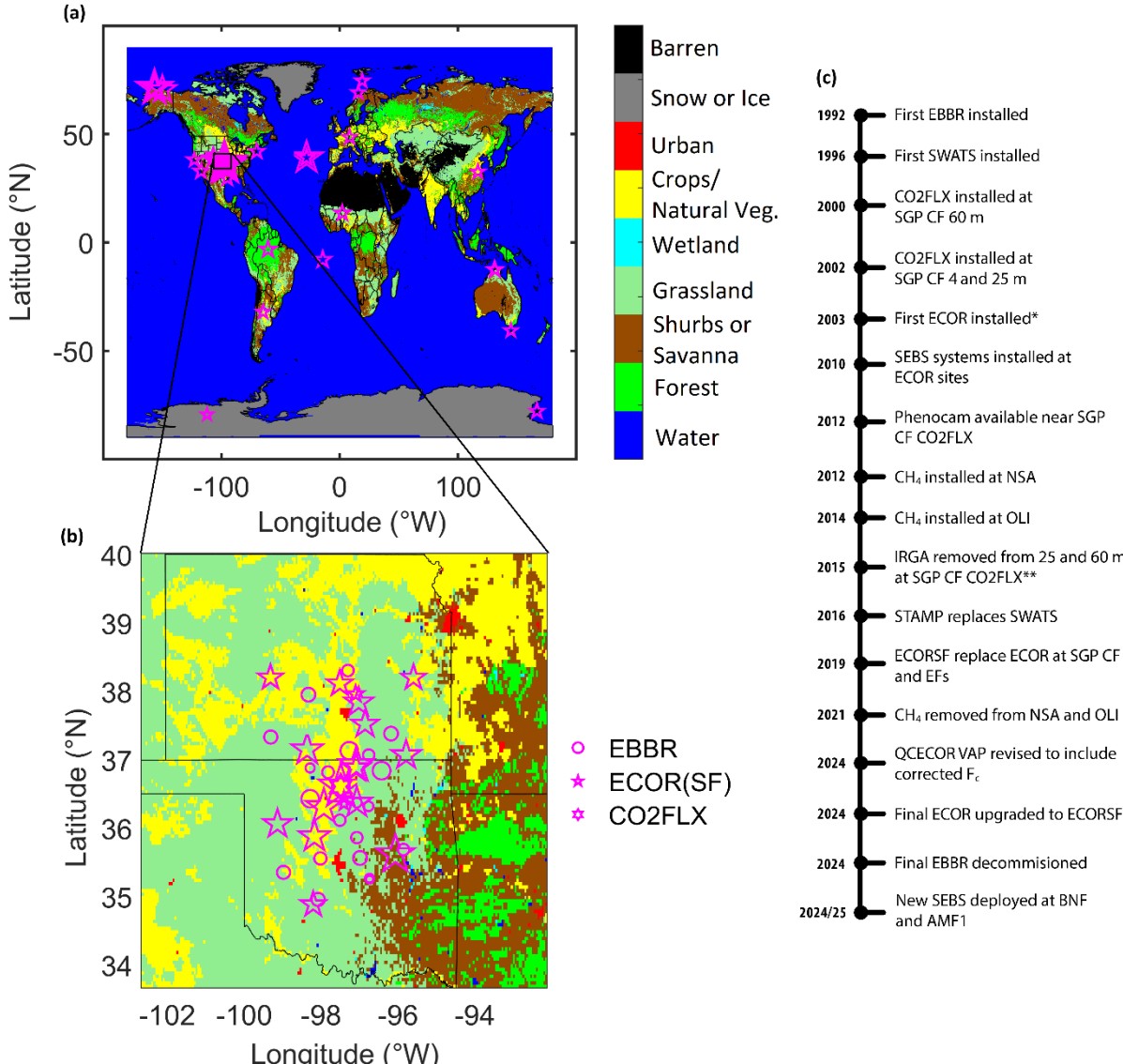

**Figure 1. (a) Global and (b) Southern Great Plains locations (magenta) of ARM EBBR (circles), ECOR/ECORSF (stars), and CO2FLX (hexagram) flux measurement systems, scaled arbitrarily by duration of deployment. Background is International Geosphere–Biosphere Programme (IGBP) land cover from combined Moderate Resolution Imaging Spectroradiometer (MODIS) Terra/Aqua 0.05 deg yearly product for 2022 (MCD12C1). (c) General timeline (not to scale) of major ARM flux instrumentation: \*First of publishable quality and quantity; data from the original ECOR (mid-1990's) is unpublished due to low quality and quantity. \*\*Note datastream name changes as described in Chan & Biraud (2022).**

The motivation for this manuscript, spurred in part from feedback from data users, is to document the ARM flux datasets in a centralized, referable format; detail data collection methods, post-processing, corrections, and best use practices; and publicize recent and planned changes to the measurement systems. The remainder of the manuscript is structured as follows: Sect. 2 documents the history of ARM flux measurements; Sect. 3 describes the datasets including their post-processing and corrections applied, additional Value-Added Products (VAPs), and general data use recommendations; Sect. 4 describes

intercomparison validation experiments between the EBBR and ECOR, and between the ECOR and external EC systems; and Sect. 5 concludes the manuscript. As the ARM user facility employs heavy use of acronyms, for ease of reading Appendix A provides a list of acronyms and abbreviations used throughout this manuscript; Appendix B contains tables describing site deployment dates, site land cover by wind direction, and soil density and texture estimates for the Southern Great Plains (SGP) facilities; and Appendix C describes ancillary datasets that may be particularly useful in scientific

analysis of the datasets described herein.

## 2 History and evolution of ARM near surface turbulent fluxes

ARM began measuring fluxes in 1992 using the EBBR system at its Southern Great Plains (SGP) site including ten grassland (mix of grazed and ungrazed) Extended Facilities (EFs; see Sect. 3.5 "Additional considerations and data use recommendations" for description of ARM naming conventions) across Oklahoma (OK, USA) and Kansas (KS, USA), and

90 at one EF (E39) installed on the northern edge of cropland, with grazed grassland to the north. The intention was that these facilities would be representative of a typical GCM grid cell and capture measurements of atmospheric processes at sub-grid cell scales in order to improve parameterization of these processes in the GCMs (Fig. 1) (Cook & Sullivan, 2025a; Stokes & Schwartz, 1994). Over the following decades, additional EFs were commissioned, while others were removed; decommissioned EFs (c. 2009/11) were primarily those at a further spatial distance from the SGP Central Facility (CF) in

Lamont, OK, and were accompanied by new EF installations closer to the CF, reflecting the evolution of increasingly higher spatial resolution of climate models (Table B1).

While the EBBR's spatial distribution provides information of sub-grid cell scale heterogeneity in sensible and latent heat fluxes, it does not provide measurements of the vertical distribution of these fluxes or fluxes of additional trace gases, such as carbon dioxide ($CO_2$). Thus in 2000, ARM commissioned the installation of the CO2FLX system at a 60 m tower at the

100 SGP CF, and shortly thereafter in 2002, additional EC systems were installed at 25 m and 4 m, on and near the tower base, respectively (Chan & Biraud, 2022). The infrared gas analyzers (for $H_2O$ and $CO_2$; IRGA) were removed from 25 m and 60 m in 2015, while the sonic anemometers remain at these heights to measure vertical profiles of H and turbulence characteristics. The forthcoming ARM Mobile Facility (AMF3) in the Bankhead National Forest (BNF; AL, USA) will include a CO2FLX system deployed at 3 heights on a 40 m tower. Observations will be conducted within and above the

105 forest canopy.

ECOR systems were established at the northern edges of crop fields (primarily wheat) at the SGP CF and eight EFs in the mid-1990s to characterize flux measurements over crops, but the data from them were of low quality and quantity during much of their usage. During 2003/4, ARM installed nine new, replacement ECOR systems across SGP at the same EFs and the CF (Cook & Sullivan, 2025b; Pekour, 2004). As with the EBBR systems, the ECOR systems' deployment locations evolved corresponding to evolving model resolution. Unlike the EBBR, the ECOR is also deployed outside of the SGP site to the North Slope of Alaska since 2011 (NSA; Utqiaġvik (formally Barrow), AK, USA), the Eastern North Atlantic since 2014 (ENA; Azores, Portugal), and three ARM Mobile Facilities (AMFs) which typically deploy for durations of approximately one (AMF1 and AMF2) to five years (AMF3) at various locations globally in response to open solicitations from the broad scientific research community (Hickmon, 2023). It is anticipated that these AMFs will continue, on an ever-roaming basis, into the future of ARM. The AMFs' ECORs have been deployed across all seven continents sampling a diversity of landscapes from rainforest to ice sheets, and marine to urban environments. Similarly to the CO2FLX system, a high-frequency $H_2O/CO_2$ IRGA is deployed, affording measurements of carbon dioxide fluxes ($F_c$) at all ECOR sites, and a methane ($CH_4$) IRGA for $CH_4$ fluxes was previously deployed at NSA (2012 – 2021) and the AMF3 deployment in Oliktok Point, AK, USA (OLI; 2014 – 2021), and will be deployed at the upcoming AMF3 deployment at BNF.

As ARM evolved following progression in programmatic and scientific needs, and in the course of streamlining instrumentation across the program, a further reduction in the number of extended facilities across SGP began in 2023. Concurrently, this marked the transition to end of operations of the EBBR systems with the final EBBR sites being replaced with ECORSF systems in 2024. The CO2FLX remains operational at SGP CF and ECOR remains in operation at SGP, NSA, ENA, and the AMFs. A list of the dates of data availability by instrument system type, and site and facility, are shown in Table B1.

## 3 Methods: Post-processing, corrections, and Value-Added Products

H and LE can be estimated using a variety of methods including lysimeters; scintillometers; flux variance, gradient, surface renewal, or bulk aerodynamic methods; energy balance models with input from numerical models or in situ measurements of meteorological and radiative states, and/or satellite-based radiance measurements; or eddy covariance, amongst others, each with varying degrees of complexity and resource constraints (Billesbach et al., 2024). Two of these methods are employed by ARM in the EBBR, CO2FLX, and ECOR instrument systems. In addition to these base datasets, ARM has developed Value Added-Products (VAPs), additional datasets that have undergone further processing for enriched scientific use, to replace flux measurements near sunrise/set with bulk aerodynamic calculations (when H and LE computed from the Bowen ratio method become nonsensical), and to apply routine eddy covariance corrections to the ECOR. These datasets are described here, and ancillary datasets that may aid in their interpretation are discussed in Appendix C.

**3.1 Energy Balance Bowen Ratio (EBBR) and Bulk Aerodynamic technique EBBR (BAEBBR)**

The EBBR measures near-surface gradients of temperature and humidity to approximate the Bowen ratio ($\beta \equiv$ ratio of sensible to latent heat flux; Eq. 1), assuming equal eddy diffusivities of water vapor and thermal heat:

$$\beta \equiv \frac{H}{LE} \approx \frac{C_P \rho}{\lambda} \frac{\overline{\Delta T}}{\overline{\Delta \rho_v}} , \tag{1}$$

where $C_p$ is the specific heat of air (J kg$^{-1}$ K$^{-1}$), $\rho$ is the density of air (kg m$^{-3}$), $\lambda$ is the latent heat of vaporization of water (or the latent heat of sublimation for frozen conditions) (J kg$^{-1}$), $\overline{\Delta T}$ is the mean temperature difference between upper and lower sensors (K), and $\overline{\Delta \rho_v}$ is the mean difference in water vapor densities between the upper and lower sensors (kg m$^{-3}$).

Gradients of temperature and humidity are measured above vegetation height using two sets of aspirated temperature and relative humidity (T/RH; Vaisala HMP45) probes mounted with a vertical separation of 1 m. Accurately measuring these

145 small gradients in temperature and atmospheric moisture is critical. However, accurate and frequent calibration of the T/RH probes is not practical, particularly across multiple, distributed sites. To overcome this challenge, the two sets of T/RH sensors are controlled by an automatic exchange mechanism, whereby the two instrument arms alternate between the top and bottom positions once each flux measurement interval (each arm is in each position 13 minutes, out of every 30 min averaging period, with a 2 minute switching period for temperature and humidity to equilibrate with ambient conditions), to

150 reduce bias or slow calibration drifts between each sensor pair.

The EBBR also measures net radiation (R; Radiation and Energy Balance Systems (REBS), Inc Q*7.1), soil heat flow (REBS HFT-3), soil temperature (REBS STP-1), and soil moisture (REBS SMP-2). The net radiometer is typically installed at 2 m and measures the sum of incoming and outgoing, long- and short-wave radiation. Surface soil heat flux, colloquially ground heat flux (G), is estimated using a suite of soil probes: soil heat flow plates are buried at 5 cm, soil moisture probes

(measuring gravimetric soil moisture) are buried at 2.5 cm to correct the heat flow measurements by accounting for the effect of soil moisture content on the soil thermal conductivity above the soil heat flow plates, and a soil temperature probe is buried across a 0-5 cm depth to estimate, along with the soil moisture measurement, energy storage between the heat flow plate and the surface. Five sets of redundant soil sensors are buried over approximately 1-2 m in the horizontal within the downward facing footprint of the radiometer to account for variability in soil properties, and the respective surface soil heat

fluxes are combined to compute an arithmetic average. The Bowen ratio is then used to partition the net available energy, approximated as net radiation less surface soil heat flux, into sensible and latent heat flux components. Summation of net radiation, surface soil heat flux, sensible heat flux, and latent heat flux thus de facto forms a closed energy budget, while additional storage (e.g. within vegetation canopy) and dissipative terms are unaccounted for.

Combining an equational form of a closed surface energy budget, where the sum of sensible and latent heat fluxes equals the

165 net radiation less energy consumed as ground heat flux (Eq. 2), and the definition of the Bowen ratio as the ratio of sensible to latent heat flux (Eq. 1 above) gives equations for the sensible (Eq. 3) and latent heat (Eq. 4) fluxes as:

$$R + G = -(H + LE + \text{other components}) , \tag{2}$$

$$H = -\frac{(R+G)}{(1+\beta^{-1})},$$ (3)

$$LE = -\frac{(R+G)}{(\beta+1)},$$ (4)

where R, G, H, and LE are in W m$^{-2}$, and "other components" are assumed to be null. These data are published as the 30EBBR datastream (Sullivan et al., 1993). Note the sign convention used in the EBBR, with negative H and LE values, as typical in daytime, indicating fluxes upward, away from the surface.

During night as the land surface experiences radiative cooling, a nocturnal inversion can form near the surface resulting in a downward sensible heat flux and negative β. As β → -1, Eqs. 3 and 4 become undefined, and H and LE become nonsensical.

This typically occurs near sunrise and sunset. Thus, ARM developed the Bulk Aerodynamic technique EBBR VAP (BAEBBR (Gaustad & Xie, 1993)), a separate datastream where, in addition to the standard 30EBBR fluxes, H and LE are also estimated using a bulk aerodynamic technique when -1.6 < β < -0.45. The BA technique computes fluxes iteratively using estimated bulk transfer coefficients for heat and water vapor that are functions of friction velocity, surface roughness, displacement height, and thus stability, and is estimated from wind speed (Met One 010C for speed and 020C for direction),

and temperature and humidity gradients for H and LE, respectively (Wesely et al., 1995).

### 3.2 Eddy Correlation (ECOR) flux measurement system, Quality-Controlled ECOR (QCECOR), and ECOR with SmartFlux (ECORSF)

Eddy covariance has been widely adapted as the gold standard method for measuring atmospheric fluxes globally across numerous networks and individual PIs, as it is one of the only methods that measures H and LE both directly and

185 independently (Baldocchi et al., 2001, 2024; Beringer et al., 2016; Chu et al., 2017; Yamamoto et al., 2005). Unlike the EBBR, in addition to H and LE, the fast response sonic anemometers and $H_2O/CO_2$ IRGAs (see Table 1 for make and model) used in the EC method afford the calculation of momentum and $CO_2$ flux across the ECOR sites. While not the primary dataset, nor focus the manuscript herein, these high frequency, raw data are archived by ARM and freely available (see Sect. C4 "Raw, fast response sonic and IRGA data"). Additionally, a methane ($CH_4$) IRGA sensor was installed at NSA

(2012 – 2021) and during the AMF3 OLI deployment (2014 – 2021) to measure $CH_4$ fluxes; this data is available in the AmeriFlux and methane VAP (AMCMETHANE; see Sect. 3.4; Billesbach, (2012)).

| | Facility | Sonic anemometer | IRGA |
|---|---|---|---|
| ECOR | SGP, ENA, AMF1 | Gill Windmaster | LI-COR LI-7500 |
| | NSA, AMF2, AMF3 | Gill Windmaster Pro | LI-COR LI-7500 |
| AMCMETHANE | NSA, OLI | Installed on ECOR | LI-COR LI-7700 |
| ECORSF | All | Gill Windmaster | LI-COR LI-7500DS |
| CO2FLX | SGP | Gill R3-50 | LI-COR LI-7500RS |
| | BNF | Campbell Scientific CSAT3B | LI-COR LI-7200 |

**Table 1. Make and model of sonic anemometers and infrared gas analysers used in the ARM EC systems. Acronyms and abbreviations used in the table are expanded in Appendix A.**

The EC method estimates fluxes from the covariance of the vertical wind speed and the quantity of interest: horizontal wind speed for momentum flux ($\tau$, Eq. 5), temperature for H (Eq. 6), water vapor concentration for LE (Eq. 7), or other scalar (e.g., $CO_2$ or $CH_4$ concentration; Eq. 8) for its respective flux:

$$\tau = \rho \overline{w'u'}, \tag{5}$$

$$H = C_p \rho \overline{w'T'}, \tag{6}$$

$$LE = \lambda \rho \overline{w'X'_v}, \tag{7}$$

$$F_C = \rho \overline{w'X'_c}, \tag{8}$$

Where w' is the instantaneous perturbation (used herein as departure of a given variable from its mean) of the vertical wind speed component (m s$^{-1}$), u' is the instantaneous perturbation of the horizontal wind speed component (m s$^{-1}$), T' is the instantaneous perturbation of temperature (K), $X'_v$ is the instantaneous perturbation of mixing ratio of water vapor in air (kg kg$^{-1}$), $X'_c$ is the instantaneous perturbation of mixing ratio of scalar "c" in air (kg kg$^{-1}$), and the overbar represents a time average operator. Note the sign convention used in the ECOR, with positive H and LE values, as typical in daytime, indicating fluxes upward, away from the surface.

Applying the eddy covariance theory in practice requires several assumptions (e.g. null mean vertical wind, no advective fluxes, steady state conditions, and that turbulence is well developed throughout the surface layer) and is subject to several instrument limitations (Foken et al., 2012). Thus prior to computing fluxes, an in-house processing code is applied to remove high frequency data spikes (Hojstrup, 1993), compensate for intrinsic time delay in the IRGA, perform a two-axis rotation such that the mean vertical and cross-stream winds are functionally nullified, and do Taylor decomposition via block averaging (Cook & Sullivan, 2025b). The de-spiked, rotated fluxes are published as the 30ECOR datastream (Sullivan et al., 1997).

Equations 7 and 8 are convenient in their simplicity. However, they are only applicable to sensors that directly measure trace gases as a mixing ratio, such as closed path sensors. When accounting for the conversion of gas concentrations measured as densities, as by open path sensors used herein, and expanding Eqs. 7 and 8, it becomes apparent that density fluctuations caused by changes in temperature or water vapor can result in apparent fluctuations in the measured trace gas of interest ($H_2O$, $CO_2$, $CH_4$, etc.) due to thermal expansion or compression, and water dilution (Foken et al., 2012). Accounting for the thermodynamic contribution of temperature fluctuations, LE can be computed as:

$$LE = (1 + \mu\sigma)\left[\overline{w'\rho'_v} + \left(\frac{\rho_v}{T}\right)\overline{w'T'}\right], \tag{9}$$

Where $\mu$ is the ratio of molar masses of dry air and water vapor, $\sigma$ is the ratio of the densities of water vapor and dry air, and T is the air temperature.

In the 1970s, Webb, Pearman, and Leuning recognized that the measured covariance between trace gas density fluctuations and vertical wind speed fluctuations were comprised of distinct components: contributions from fluctuations in temperature, water vapor, atmospheric pressure, and other trace gases (Lee and Massman 2011). Only one of these was caused by the vertical transport of trace gas of interest, which is the desired outcome of the measurement. The others were either

thermodynamic effects on the atmosphere, or the confounding effect of the simultaneous transport of water vapor (confounding effects of other trace gas transport, while present, are generally small and ignored). Of the two thermodynamic components that are related to fluctuations of temperature and pressure, only the temperature component is large. Except in a few extreme cases of high elevation locations, the pressure fluctuations can be ignored. This leaves the sum of three terms that make up the measured covariance. To obtain the true flux of the trace gas of interest, we must subtract the temperature and water vapor fluctuation terms from the measured covariance.

$$F_c = \overline{w'\rho'_c} + \mu\left(\frac{\overline{\rho_c}}{\overline{\rho}}\right)\overline{w'\rho'_v} + (1 + \mu\sigma)\left(\frac{\overline{\rho_c}}{\overline{T}}\right)\overline{w'T'}, \tag{10}$$

Where $\rho_c$ is the density of scalar "c" (kg m$^{-3}$). These apparent fluxes are corrected by including these additional Webb-Pearman-Leuning, or "WPL", correction terms (Eqs. 9 and 10; Webb et al., 1980). For Eq. 10, the first term is the measured covariance, the second term is the contribution from the vertical transfer of water vapor, and the last term is the thermodynamic contribution of temperature fluctuations. In practice, all of these terms must be accounted for when an open-path IRGA, such as the LI-7500* series, is used. When closed-path or "enclosed" path (e.g., the LI-COR LI-7200 on the CO2FLX at BNF) instruments are used, it has been shown that the last term (thermodynamic or temperature term) becomes negligible, and only the first two terms need be considered. It's important to note that the covariances contained in the second and third term should be fully corrected for frequency effects, as discussed next. Under most conditions, the last term (thermodynamic or temperature) is usually larger than the second (water vapor).

In the EC method, several instrument limitations and post-processing methods act in practicality as low- and high-pass filters (Burba & Anderson, 2010) to the computed fluxes, for which various analytical and empirical spectral correction methods have been proposed to account for this frequency attenuation (W. Massman & Clement, 2004). As with the EBBR, a VAP was developed to account for the above necessary eddy covariance corrections: the Quality-Controlled Eddy Correlation (QCECOR (Gaustad, 2003)) flux VAP (Tao et al., 2024). Prior to the addition of the WPL terms (Eqs. 9 and 10; Webb et al., 1980), the VAP corrects for frequency attenuation resulting from sensor separation (between the sonic and IRGA), stability, and path-length and volume averaging (Andreas, 1981; Kaimal, 1968; Kristensen & Fitzjarrald, 1984; W. J. Massman, 2000). Further, quality control steps are applied to the ECOR data to remove suspicious data points: this includes removing data outside minimum and maximum thresholds (H and LE > |150| W m$^{-2}$ during the night, and H and LE < -100 W m$^{-2}$ when solar insolation is > 300 W m$^{-2}$), removing outliers falling outside of four standard deviations from the diurnal or nocturnal mean, and a temporal stability check is applied over a moving window of ± 3 hours (Tao et al., 2024).

The original QCECOR VAP, as documented in Tang et al. (2019b), also removed data as incorrect when a co-located wetness sensor indicated the potential for water (such as precipitation, dew, or frost) on the IRGA sensor optical path (part of the Surface Energy Balance System (SEBS) installed at ECOR sites beginning in 2010; see Appendix C1 for details). However, in 2024, the QCECOR VAP was modified to no longer remove data explicitly based on measurements from the wetness sensor; alternately, the wetness variable is included as an additional variable in the QCECOR to aid data users in interpretation of the flux data and identification of periods when the fluxes may be considered suspect. At the same time, the

QCECOR code was also modified to apply the aforementioned corrections to $F_c$ (c.f. only to H and LE in the original release). These modifications are currently in production and will be applied retroactively to all past and to all forthcoming QCECOR data (Tao et al., 2024).

The ECOR remains in operation at SGP, NSA, ENA, and the AMFs. However, the ECOR system itself has not remained static. Due to sensors becoming obsolete (i.e. parts no longer supplied or serviced by vendors), an upgrade to the ECOR

systems was proposed in 2018, implemented at SGP in 2019, and completed a progressive rollout across all ARM ECOR installations in late 2024. The new design was equivalent to the existing system, with newer model sonic anemometers (mix of Gill Windmaster and Windmaster Pro vs. Gill Windmasters in the old and new systems, respectively) and IRGAs (LI-COR LI-7500 vs. LI-7500DS in the old and new systems, respectively; Table 1); unlike the original ECOR which computed the fluxes using in-house code and required a VAP, QCECOR, to post-process the fluxes with routine eddy covariance flux

corrections, the new systems include on-board microprocessors (SmartFlux 3, LI-COR Biosciences) for computing both raw and corrected fluxes using the EddyPro software (LI-COR Biosciences, 2021). The new generation of ECOR is therefore designated ECORSF (ECOR with SmartFlux; Sullivan et al., (2019a)).

To correct fluxes from the ECORSF systems, EddyPro was run in express mode. As all of the Gill Windmaster sonic anemometers were purchased after identification and correction of the "w-boost" bug (Billesbach et al., 2019), no fix was

necessary, nor was the angle of attack correction applied. As with the ECOR and QCECOR post-processing, EddyPro applies a two-axis rotation of the sonic anemometer wind measurements, block averaging for Taylor decomposition of the time series, WPL terms to compensate for density fluctuations, and accounts for sensor time lags using the covariance maximization method. In addition to standard ARM QC flagging on data based on valid minimum and maximum values (30ECOR/ECORSF variable field "qc_[variable_name]"), EddyPro employs additional quality control procedures, with

results available in output datafiles. This includes tests for steady state conditions and well developed turbulence, following the 0 ("best quality fluxes") – 1 ("suitable for general analysis such as annual budgets") – 2 ("fluxes should be discarded") system of Mauder & Foken (2015) (30ECORSF variable field "flag_[variable_name]"), and flags for tests of spikes, amplitude resolution, drop outs, absolute limits, and skewness and kurtosis in the data (LI-COR Biosciences, 2021).

**3.3 Carbon Dioxide Flux (CO2FLX) measurement system**

The CO2FLX datastream comprises a number of instrument packages, primarily located at the ARM SGP CF. Similar to the ECOR systems, the CO2FLX quantifies turbulent fluxes using the eddy covariance technique. The CO2FLX also includes a full complement of meteorological (Koontz et al., 2016b), below-ground (Koontz et al., 2015d), and radiation (Koontz et al., 2016a) observations (see Sect. 6.2. "AmeriFlux Measurement Component (AMC)"). From 2002 – 2015, $CO_2$ and $H_2O$ fluxes were collected at three heights (4, 25, and 60 m). In 2015 the infrared gas analyzers were removed from 25 and 60 m. The

current 4, 25, and 60 m datastreams (Koontz et al., a-c) include turbulent statistics and fluxes of momentum and sensible heat from a Gill R3-50 sonic anemometer, while the current 4 m flux datastream (Koontz et al., 2015a) also includes $CO_2$ and $H_2O$ fluxes from an infrared-gas analyzer (LI-COR LI-7500RS).

The eddy covariance processing for the 4, 25, and 60 m are performed on a daily basis by the ARM Data Center using EddyPro in advanced mode, where spectral corrections from Massman (2000, 2001) were applied (c.f., Moncrieff et al. (1997) in express mode), and the default lag settings are also adjusted to account for fixed lags introduced by the data acquisition system.

The upcoming AMF3 deployment in the Bankhead National Forest (BNF) will include three heights of $CO_2$ and $H_2O$ fluxes along a 40 m tower. The highest level will also include instrumentation for $CH_4$ flux observations. The primary eddy covariance sensors at AMF3 will differ from those at SGP: A Campbell Scientific CSAT3B sonic anemometer will be used rather than the Gill R3-50 and an enclosed-path LI-7200 infrared gas analyzer will be deployed (Table 1). EddyPro configurations will be similar to CO2FLX at SGP.

The integrated CO2FLX dataset from SGP is also contributed to the AmeriFlux network under the site identifier US-ARM and the full record can be accessed in two forms: the AmeriFlux BASE data product (Biraud et al., 2024) contains the quality controlled, half-hour fluxes (all heights) and ancillary observations; the AmeriFlux FLUXNET data product (Biraud et al., 2022) includes gap-filled and partitioned fluxes that are produced using ONEFlux code (Pastorello et al., 2020).

### 3.4 AmeriFlux and Methane (AMCMETHANE) VAP

As discussed above, a $CH_4$ IRGA was previously deployed on the ECORs at NSA (2012 – 2021) and OLI (2014 – 2021), and is published as the AmeriFlux and Methane (AMCMETHANE) VAP (Billesbach, 2012). Since the NSA and OLI methane flux systems pre-date EddyPro, a set of in-house programs were used to process and quality control the AMCMETHANE VAP. This suite of software was used by the AmeriFlux program to validate the results from EddyPro processing prior to the adaptation of that program for their standard data post processing. The basic scheme was the same as detailed above. In addition, as required for single-line absorption measurements, as made by Tunable Diode Laser Spectrometry (TDLS), a set of spectral line corrections were applied to the methane fluxes. Raw data from three separate instrument systems (ECOR, SEBS, and AMC) were combined, and processed by the suite of programs mentioned above to produce a master data file with 30-minute averages, fluxes, and estimated flux uncertainties (Billesbach, 2011). This master data file was then further processed by another program to evaluate and attach QA/QC codes, and to output files formatted for inclusion in the ARM (Billesbach, 2012) and AmeriFlux (OLI: US-A03 (Billesbach & Sullivan, 2020a, Sullivan et al., 2025a) and NSA: US-A10 (Billesbach & Sullivan, 2020b, Sullivan et al., 2025b)) archives on an annual basis.

Located in the Arctic, both NSA and the former OLI sites are subject to harsh environmental conditions. Additionally, both sites are coastal and thus prone to a buildup of sea salt on the sensors' optics. However, due to local regulations, routine use of mirror washing fluid was not an option. To account for these limitations, a quality control procedure was implemented where data were flagged as bad when the $CH_4$ reference signal strength fell below a threshold of 10 %. While this threshold is very low (c.f., a typical reference signal strength of 40 – 60 %), and in other environments would not be considered acceptable, it was necessary for these harsh conditions. The lower value adds more noise and uncertainty to the measurements and must be considered when analyzing this data.

## 3.5 Additional considerations and data use recommendations

ARM data described herein are stored in the standardized NetCDF format, for which programming interfaces are readily available within numerous, commonly used languages (NSF Unidata 2025). One such interface, developed for use in a Python environment, is the Atmospheric data Community Toolkit (ACT). ACT is an open-source Python library designed to simplify the analysis and visualization of atmospheric data (Theisen et al., 2024). It was developed to assist researchers in accessing, processing, and interpreting data from various sources, particularly ARM's extensive archive of atmospheric observations. ACT supports reading multiple data formats, such as NetCDF, commonly used by ARM, and provides tools for applying additional quality control. ACT also includes a variety of utilities for visualization, retrievals, corrections, and more (https://github.com/ARM-DOE/ACT). Documentation for ACT is available at https://arm-doe.github.io/ACT/, including a general user guide with information from installation to usage, an API reference manual outlining available functions, and a gallery of example workflows.

Through Data Discovery (https://adc.arm.gov/discovery/#/), ARM's primary interface for data distribution, data users can query data by instrument datastream, specific site or field campaign, and/or by date, amongst other search parameters. For users interested in automating downloading specific datastreams, the ARM Live Data Web Service (https://armlive.svcs.arm.gov/) was developed to allow access to URL based download links, outline Wget and cURL command usage, and provide example scripts for automated data access. Software for querying this web service is also available in Python through ACT.

When using ARM flux data from the systems described herein (EBBR, ECOR/ECORSF, and CO2FLX) it is recommended to:

- Use fully corrected fluxes (from the VAPs BAEBBR and QCECOR, and "corrected_[variable name]" in ECORSF). For preservation of data provenance, these VAPs are published as additional datastreams to the standard base products; e.g., the 30ECOR datastream includes 30-min, de-spiked and rotated, but otherwise uncorrected fluxes, while the 30QCECOR datastream includes the 30-min fluxes computed with the routine eddy covariance corrections, described in Section 3.2, applied, in addition to the uncorrected fluxes, and the 30EBBR datastream includes 30-min fluxes, as described in Section 3.1, while the 30BAEBBR datastream includes the additional flux variables, as computed from the bulk aerodynamic calculations, in addition to the 30EBBR fluxes.
- Use caution when interpreting data when fetch is inadequate (see Sect. 3.5.1 and Tables B2-B4).
- Use embedded quality control ("qc_[variable name]", all datastreams) variables and EddyPro flags ("flag_[variable name]", ECORSF only) to filter out potentially erroneous data.
- Consider, and disregard data as appropriate, following recommendations from Data Quality Reports (DQRs) for known issues not characterized by embedded qc variables. These reports are available from https://app0.arm.gov/dqr/#s/_r::_. As noted above, ACT provides an example interface for interacting with ARM data, including querying the DQR database (https://dqr-web-service.svcs.arm.gov/docs) through the "qc" function and "add_dqr_to_qc" subfunction (See

https://arm-doe.github.io/ACT/source/auto_examples/qc/plot_dqr_qc.html#sphx-glr-source-auto-examples-qc-plot-dqr-qc-py for an example workflow).

And be aware:

- Preventative maintenance is performed bi-weekly. During these times, general inspection of the instruments is performed and sensor heads (sonic, IRGA, radiometers, rain detector/wetness) are cleaned. ECOR and CO2FLX IRGAs are scheduled to be calibrated annually.

- Time stamps are at the beginning of the half hour for the ECOR and CO2FLX, but at the end of the half hour for the ECORSF, SEBS, and EBBR.

- For the ECOR and CO2FLX, positive values indicate fluxes away from the surface (typically upward/positive and downward/negative flux of H and LE, and $CO_2$, respectively, during daytime), SEBS positive values indicate fluxes toward the soil surface (typically downward/positive net radiation and downward/negative surface soil heat flux during daytime), and EBBR negative values indicate fluxes away from the surface (typically upward/negative fluxes of H and LE and downward/negative surface soil heat flux during daytime).

- Gas concentrations, and thus LE and $F_c$, from the ECOR and CO2FLXs' IRGAs may be erroneous during precipitation, fog, or dew/frost. Beginning in 2010, ARM installed Surface Energy Balance Systems at all ECOR sites. While these systems are intended to provide radiative and surface soil heat fluxes to complement the turbulent fluxes, they also include a wetness sensor that provides a qualitative assessment of the potential presence of water on the sensors (see Appendix C1 "Surface Energy Balance System (SEBS)"). Additionally, for the newer ECORs ("ECORSF"), a $CO_2$ signal strength variable is useful in identifying when the IRGA optical path is potentially obstructed, and the CO2FLX data streams include a qc flag for low signal strength.

- The naming convention for ARM instrument locations include an observatory name (e.g. SGP = Southern Great Plain) indicating the specific site or campaign and a qualifier for the specific facility where the instrument is located within that observatory (B = Boundary Facility, C = Central Facility, E = External Facility, I = Intermediate Facility, L = Logistics Facility, N = Network Location, S = Supplemental Facility, or X = External Data / Facility, followed by a unique number to that specific facility)

- For ARM data, the naming convention is: [site identifier][duration][abbreviated instrument name][specific data set produced by instrument, optional][facility].[data processing level].[date.time].[file type]. E.g. the processed ("b1") NetCDF ("cdf") 30-min ("30") ECOR ("ecor") at the Barrow, AK extended facility ("E10") at the North Slope of Alaska ("NSA") site on 4 July 2017 ("20170704.000000") is "nsa30ecorE10.b1.20170704.000000.cdf".

- Several known environmental or instrument issues impact data on a reoccurring basis, including: frozen or otherwise obstructed sensor/hardware, particularly the EBBR automatic exchange mechanism; damage to radiometer domes from bird claws; damage to soil sensors caused by wildlife; sensor and hardware failure; or power outages. These periods are documented in Data Quality Reports (DQRs) when data is impacted and identified.

- There are numerous data streams containing the name CO2FLX and the instrument handbook (Chan & Biraud, 2022) is helpful to identify and differentiate them. Note that the data stream names changed in 2015.

### 3.5.1 Fetch and dependence on wind direction

While the measurements from the EBBR, CO2FLX, and ECOR are physically point observations, by averaging (a theoretical requisite of the methods) over 30-minute intervals, the measurements are reflective of the air masses' interaction with the surface over which the transient eddies transverse during the sampling interval, referred to as the fetch or flux footprint (Chu et al., 2021). Thus, the ideal measurement site would be surrounded by a landscape with homogeneous surface characteristics (vegetation and soil conditions, surface roughness) and minimal obstructions (building, trees in a non-forest site, structures
from other instruments). However, this is challenging in practice; thus, consideration of prevailing wind direction during a given measurement interval, and consequently the landscape being "seen" by the sensor, is necessary to properly interpret the measured flux values.

Being an atmospheric observatory, ARM does not routinely publish comprehensive, site specific or temporally variant vegetation characteristics. However, acceptable wind directions for the SGP CF and EFs, and a rough estimate of the
405 vegetation type within the fetch footprint of the ECORs and EBBRs are given in the respective instrument handbooks (Cook & Sullivan, 2025a,b) and reproduced here in Tables B2-B4. These data are compiled from a combination of on-site observations during installation or site visits, site technician reports, and maps and satellite-based imagery. Common crops across SGP include winter wheat, soybeans, alfalfa, sorghum, and corn, but the specific crop planted varies season by season (Raz-Yaseef et al., 2015), and double-cropping is not uncommon. For a qualitative assessment of temporal phenology of the
410 vegetation at a specific site, it is recommended that data users consult external datasets, such as vegetation indices from satellite-based sensors (e.g., from Landsat, Sentinel, MODIS, VIIRS) or other vegetation synthesis databases (e.g., United States Department of Agriculture's CropScape). Since 2002, visible and infrared imagery have been taken at the ARM CF crop field near the CO2FLX tower and are available through the PhenoCam network sites "armoklahoma" (2002 – 2014) and "southerngreatplains" (2012 – present) (Seyednasrollah et al., 2018).

## 4 Results from intercomparison experiments

Users of data sets invariably look for or assume certain assurances about that data. These include accuracy, precision, and consistency, with the latter often being defined as traceability to standards. For many instruments and measurements, this is achieved through regular comparisons to standards or calibrations. For other measurements, this is not possible because standards simply do not exist. In these cases, intercomparison of many measurement systems to a single, well vetted system
is often substituted. The ARM ECOR and EBBR systems both fall into this category. Both systems measure fluxes of energy and atmospheric trace gases for which no standards can exist. Other networks (e.g., AmeriFlux, NEON) have adopted the intercomparison approach to validate their flux products and to provide a network-wide quality standard for their

instrument systems and flux data products (Schmidt et al., 2012). Accordingly, benchmarking experiments were conducted to provide this type of data product validation, and to link the flux products from the EBBR, ECOR, and CO2FLX systems:

comparison of a pair of co-located ARM ECORSF and EBBR systems, and comparisons of the ARM ECORSF and CO2FLX with external EC systems (an AmeriFlux site, and an independently designed portable, roving system). The intent of these experiments is thus, not to fully characterize these datasets, but to demonstrate data quality in terms of self-consistency, or lack thereof, between the various measurement systems.

## 4.1 Intercomparison of EBBR vs. ECORSF

The energy balance Bowen ratio and eddy covariance methods both measure H and LE, and thus data acquired by the two methods are, at least superficially, equivalent. However, the two techniques operate on different theoretical principles and assumptions, e.g. EBBR assumes, by definition, a closed energy balance, while failure to close the energy balance is a well-documented phenomena in EC research (Twine et al., 2000); thus, perfect agreement between the EBBR and EC systems' measurements is not expected, even in ideal environmental conditions (Billesbach et al., 2024). Further, even when within

the same vegetation field, the actual footprints being measured by each system are not exactly the same, with the ECOR fetch generally extending to a greater distance from its sensors than for the EBBR (Cook & Sullivan, 2025a,b).

As discussed above, in general across the SGP, the EBBR were deployed within grassland and the ECOR were deployed on the northern edge of crop fields. Thus, while ARM has been making measurements to be representative of both grassland and crop fluxes, interpretation of these datasets to characterize the impact of vegetation type on near surface turbulent fluxes is

confounded by the differing underlying instrument method used to acquire these datasets. This was briefly addressed in Bagley et al. (2017), where an eddy covariance system was collocated with the EBBR at E32 for 8 months in 2016, and the observed fluxes were compared. They found high instantaneous agreement ($R^2 = 0.79$ and 0.73 for H and LE, respectively) and a low bias (regression line within 3% of a 1:1 line) between the two instrument methods during midday and concluded that the instrument effects on fluxes are small and data from the two systems were suitable for use in a synthesis analysis.

To further address the potential discrepancy between measurements obtained using the two methods and further understand the impact of the two instrument types, vs. the impact of local vegetation type, when spatially averaging these datasets, Tang et al. (2019a) compared the pseudo co-located EBBR and ECOR at the SGP CF, one of two locations at which ARM has historically deployed both an ECOR and an EBBR simultaneously. While in close proximity to each other (within a few 100 m's), interpretation of their data comparison was restricted due to differing vegetation within the flux footprints of the

respective systems. In addition to the SGP CF, an EC system (ECOR through Oct 2019, ECORSF thereafter) and an EBBR system were also co-located at SGP E39 from 2015 – 2023; unlike the CF, the EC and EBBR at E39 were only separated by a few meters and measure fluxes from within the same approximate fetch footprints – crop (typically winter wheat) to the south (~100 – 260º) and ungrazed grass to the north (~0 – 80º and 280 – 360º). However, when Tang et al. (2019a) conducted their research, it was determined that the duration of data from E39 was insufficient for a robust analysis and thus

the site was excluded from their analysis. With a longer data record now available, herein we extend the work of Tang et al. (2019a) to include data from the co-located flux systems at E39.

To facilitate the comparison, the data were divided into subsets based on vegetation conditions: periods of southerly winds with fetch over cropland vs. northerly winds with fetch over grassland.  For the comparison, only periods with data available for both the EC and EBBR systems were considered, periods where either system had quality control flags not equal to zero

or a Data Quality Report (DQR; Sect. 3.5) indicating incorrect data were removed, and only corrected (QCECOR VAP for ECOR, "corrected_[flux variable]" for ECORSF, and BAEBBR) flux data were considered.

As anticipated from previous literature, a stronger agreement between the ECOR and EBBR was measured for H than for LE: averaged over all conditions, Pearson's linear correlation coefficients ($\rho_P$) = 0.94 and 0.89 and biases = 1.0 and 50.8 % (as quantified by the deviation of the orthogonal linear regression slope from unity, with > 0 indicating |EBBR| > |ECOR|)

for H and LE, respectively (Fig. 2a,b; Table 2). This discrepancy is apparent when focusing on the typical diel cycle in heat fluxes, with a maximum difference in hourly means measured by the EBBR and EC systems occurring around 13 – 14 LST of 12 W m$^{-2}$ and 69 W m$^{-2}$, for H and LE, respectively.  The difference in H is unchanged when considering only data from fetch over crop vs over grass (Fig. 2e). Conversely, the disagreement is larger for LE when fetch is over crop (82 W m$^{-2}$) than over grass (50 W m$^{-2}$); however, these differences should be viewed in the context that there is substantial overlap in the

470 day-to-day variability in the two distributions, as demonstrated by the overlap in their hourly standard deviations (Fig. 2c,d).

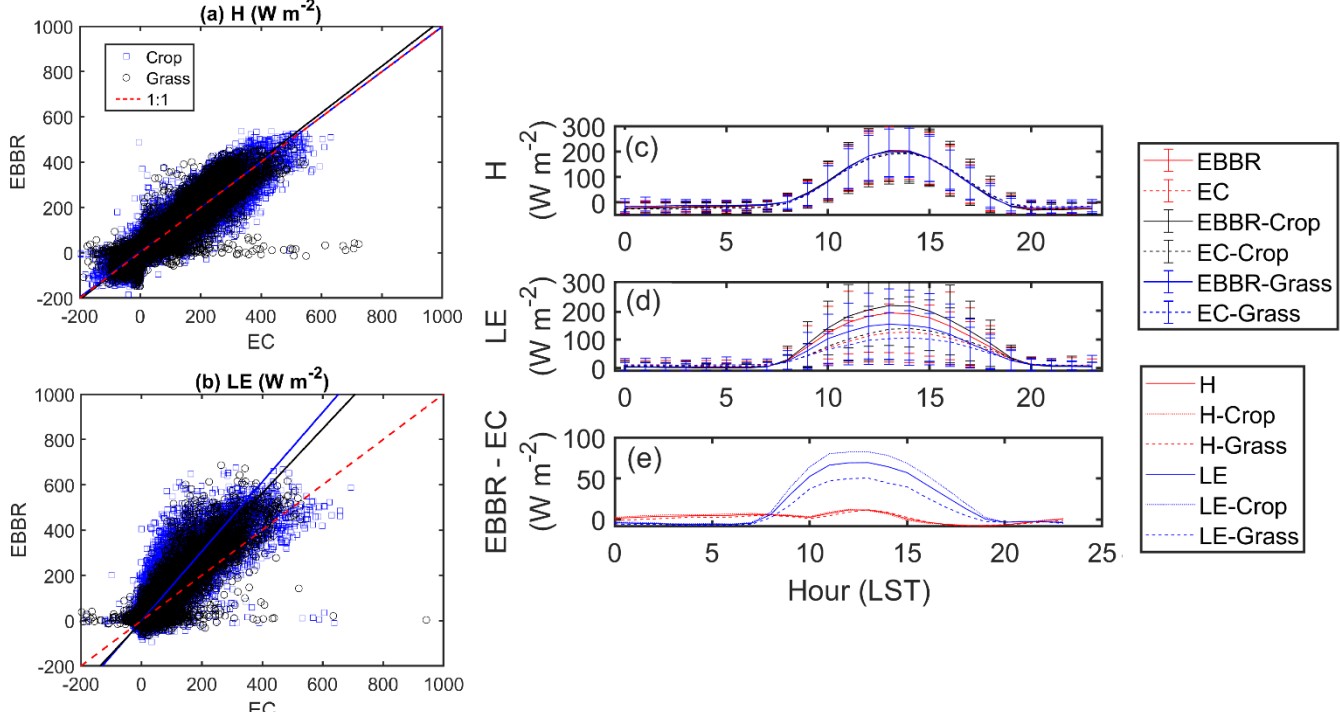

**Figure 2. Scatterplot comparison of sensible (a) and latent (b) heat fluxes from 2015 – 2023 at SGP E39. Data are segregated by prevailing wind direction and resultant vegetation type within the measurement footprint: southerly (100 – 260°) wind and crop (blue squares), and northerly (0 – 80° and 280 – 360°) wind and ungrazed grass (black circles). Also shown is a 1:1 line for reference (red, dashed) and orthogonal linear regression lines for crop (blue) and grass (black). Mean (line) and standard deviation (whiskers) H and LE, and the mean EBBR – EC difference in diel cycles are shown in (c), (d), and (e) respectively, segregated by all wind directions, and when fetch is over crop vs. grass. Note, as the sign convention differs between the EBBR and EC, all EBBR fluxes were multiplied by -1.**

| H | | | |
|---|---|---|---|
| | All | Crop | Grass |
| $\rho_P$ | 0.94 | 0.94 | 0.93 |
| Bias (%) | 1 | -0.4 | 3.1 |
| LE | | | |
| $\rho_P$ | 0.89 | 0.89 | 0.9 |
| Bias (%) | 50.8 | 53.6 | 42.2 |

**Table 2. Pearson's linear correlation coefficient ($\rho_P$) and bias (quantified using the deviation of the orthogonal linear regression slope from unity, with > 0 indicating |EBBR| > |ECOR|) for the intercomparison between the co-located EBBR (ordinate) and ECOR (abscissa) at E39 from 2015 – 2023. Statistics are also subset by vegetation within the flux footprint, as determined by prevailing wind direction, with crop (typically wheat) to the south (~100 – 260°) and ungrazed grass to the north (~0 – 80° and 280 – 360°).**

These findings are supportive of the conclusions presented by Tang et al. (2019a) that, on average, LE measured from the EBBR was greater than from the EC systems. In their study, Tang et al. (2019a) attributed the differences between the EBBR and ECOR, in part, to differences in vegetation upwind of the two systems. Specifically, when the datasets were segregated

by wind direction, the observed differences were significant when the upwind fetch differed between the two systems, but while nonnegligible difference were also observed when both systems had upwind fetch over the same vegetation (grass), they were no longer statistically significant. No clear dependence of the agreement on vegetation type was observed at E39, with comparable disagreement in LE with fetch over crop and over grass. Given that the spatial separation between the systems at the CF is much larger (100s of m's) than at E39 (a few m's) and heterogeneity in vegetation at CF is greater than at E39, even when classified by predominant vegetation (i.e., obstruction or interference from more ancillary instruments and vegetation management at CF, particularly in the field in which the EBBR was deployed; see Tang et al. 2019a's Fig. 1), we conclude that the differences between LE measured at E39 by the two methods (EBBR and EC) are reflective of differences in the instrument systems themselves, not solely due to environmental factors. As with the findings of Billesbach et al. (2024), this analysis underscores that larger instantaneous uncertainty exists for individual measurements, particularly for LE.

## 4.2 Intercomparison of ECORSF vs. AmeriFlux

Fifteen years after the 2003/2004 ARM ECOR installations across the SGP, degradation of the sonic anemometers and IRGAs became increasingly prevalent, and the instrument vendors had ceased manufacturing the existing models, declaring them obsolete and no longer eligible for service and repairs. As deployed sensors failed, spare sensors dwindled, and requisition of newer models was needed. As sensor technology, and the field of eddy covariance measurements in general, had greatly evolved over the prior decade and a half since the inception of the ECOR, rather than retrofit newer model sensors to the 2003/2004 ECOR system design, it was elected to conduct a complete overhaul of the ECOR systems (ECORSF, Sect. 3.2). Although side by side comparison between each old and new ECOR system was not logistically feasible, two intercomparison validation exercises were conducted. A similar comparison was also previously performed at the SGP CO2FLX as part of its inclusion in the AmeriFlux network and is briefly revisited here.

### 4.2.1 Comparison of CO2FLX with AmeriFlux portable eddy covariance system

In 2006 and 2015, the AmeriFlux project technical teams conducted inter-comparison experiments at the SGP CF, deploying a portable eddy covariance system (PECS, Billesbach et al., 2004) side by side the CO2FLX system for 1 to 2 weeks. These exercises were led by Oregon State University and Lawrence Berkeley National Laboratory personnel. Results from two exercises showed that comparison of sensible and latent heat, and carbon fluxes between the in situ and PECS systems were within 10 % of each other, or within measurements uncertainties.

### 4.2.2 Comparison with AmeriFlux site US-IB2: Fermi National Accelerator Laboratory – Batavia (Prairie site)

Shortly after the 2003/2004 ECOR installations across SGP, an additional set of EC sites were established on the Fermi National Accelerator Laboratory (Fermilab) campus, in Batavia, Illinois as part of the U.S. DOE AmeriFlux network. The flux systems were designed and operated by the ARM instrument mentor and have the same components and specifications

as the 2003/2004 ECOR systems. The consistency between the ARM ECOR and the Fermilab EC system design, and the proximity of Fermilab to the ARM ECOR mentors' home institution, provided an ideal opportunity to co-locate and intercompare the new ECORSF with an EC system analogous to the 2003/2004 ECOR system.

After development of an ECORSF prototype, it was deployed a few meters from the US-IB2 flux site (Matamala, 2019) for two months (July and Aug 2018). The site is located in the middle of a restored prairie, with adequate fetch in all directions except for due east. For the comparison, fully corrected fluxes were used, and only high-quality fluxes were considered (qc flags = 0). Following site operator recommendations, AmeriFlux data were further filtered to remove LE fluxes when they were < - 25 W m$^{-2}$ (downward) during the day, when the $CO_2$ fluxes were flagged as bad, and when $CO_2$ fluxes that were positive (upward) during the day. Of the 2877 half-hours (~ 60 days) of measurements, this QA/QC procedure left ~ 65 – 70 % of the flux data, depending on specific variables.

The two collocated flux systems exhibited considerable agreement. All fluxes had Pearson's linear correlation coefficient ($\rho_P$) between 0.95 and 0.97, with the lowest agreement for $F_c$, and highest agreement for H, with LE and friction velocity ($u_*$) agreement middling (Fig. 3; Table 3). Similarly, the bias (as quantified by the deviation of the orthogonal linear regression slope from unity, with > 0 indicating |US-IB2| > |ECORSF|) was only 2.8 % for H, 4.6 % for $u_*$, -18.4 % for $F_c$, and -8.4 % for LE. It is noteworthy that the magnitude of the daytime LE and $F_c$ are larger (more positive/upward and more negative/downward for LE and $F_c$, respectively) from ECORSF than from US-IB2, potentially due to increased $H_2O$ and $CO_2$ precision of the newer LI-7500DS in ECORSF c.f. the older LI-7500 used in the US-IB2 system (Fig. 3). However, the larger bias for LE and $F_c$ is also consistent with the degree of heterogeneity in vegetation density and species even over the small spatial separation of the two flux systems (~ 5 m), and nighttime $F_c$ (respiration) was larger in the ECORSF measurements. Thus, at least some of the differences may be driven by slight variability in vegetation within the respective flux footprint of the two systems.

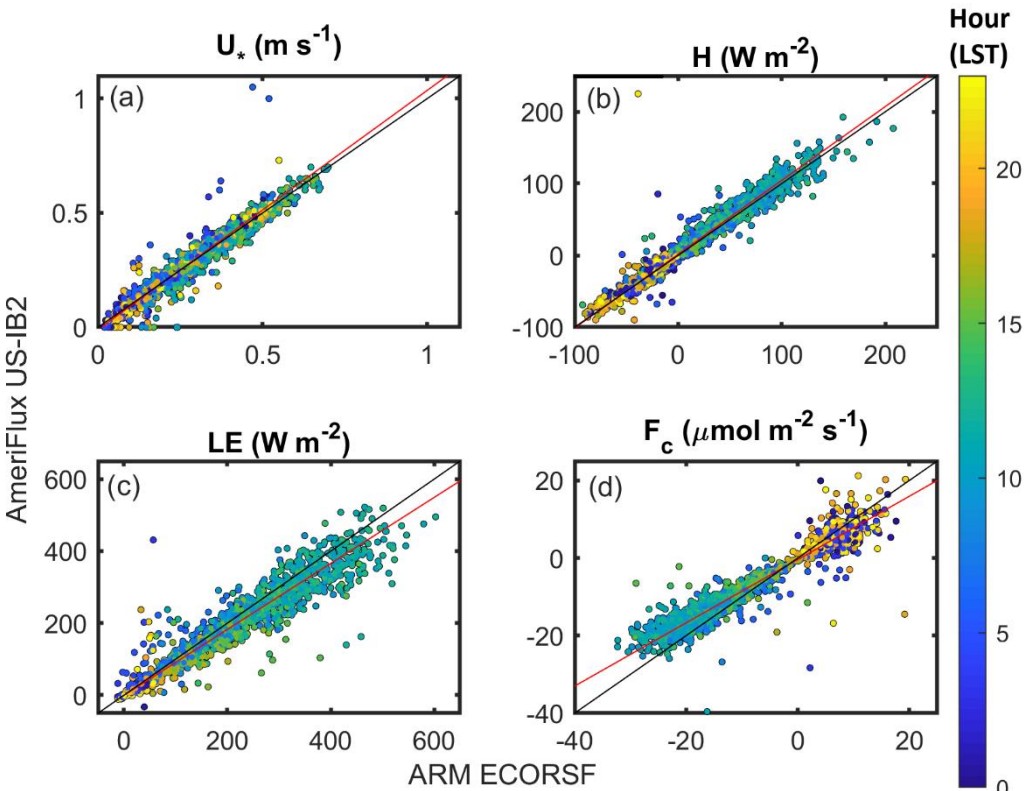

**Figure 3. Scatterplots of fluxes from the ARM ECORSF prototype (abscissa) and AmeriFlux US-IB2 Fermilab prairie (ordinate) for July and August 2018. Color scale indicates the hour of the measurement in local standard time, 1:1 lines are shown in black, and orthogonal linear regression lines are in red.**

|  | H | LE | $F_c$ | $u_*$ |
|---|---|---|---|---|
| $\rho_P$ | 0.97 | 0.96 | 0.95 | 0.96 |
| Bias (%) | 2.8 | -8.4 | -18.4 | 4.6545 |

**Table 3. Pearson's linear correlation coefficient ($\rho_P$) and bias (quantified using the deviation of the orthogonal linear regression slope from unity, with > 0 indicating |US-IB2| > |ECORSF|) for the intercomparison between the co-located US-IB2 (ordinate) and ECORSF (abscissa) EC systems at Fermilab July and Aug 2019.**

### 4.2.3 Comparison of ECORSF with roving, AmeriFlux[-like] portable eddy covariance system

During a 2008 ARM Cloud Modeling Working Group meeting, it was proposed to run an intercomparison validation experiment with the ARM ECOR systems. The concept was well received, but timing and funding for the proposed project were deficient, and the concept was put on hold indefinitely. Nearly a decade later, while upgrading the ECOR system with what would become the ECORSF system, the concept of the validation experiment was resurrected in 2018, and funded to proceed. A portable EC system was designed and built, in a comparable fashion to the AmeriFlux PECS (Billesbach et al.,

2004), shortly thereafter.

In this campaign, the EC validation ("reference") system was set up at each of the ARM SGP ECORSF sites. Raw data were collected for a period of 1 to 2 weeks, with the validation system installed 3 to 5 m east of the ECORSF tower in all cases, and the validation instruments adjusted to approximately the same height above ground as the corresponding ones on the ECORSF tower. This arrangement was chosen to keep the footprints, as seen by both sets of instruments, as similar as possible, while avoiding any potential interference between the systems. The raw data from the validation system were acquired and processed with the HuskerProc program and compared to the published ARM ECORSF data. To eliminate any potential bias due to different QA/QC procedures, and to maximize the amount of data available for comparison, a single set of valid maximum and minimum values were applied to both data sets. Because conditions at each site were unique (environmental and growth stage), the actual maximum and minimum values were adjusted for each site, through trial and error, to eliminate obvious, extreme outliers and non-physical values.

Intercomparisons were performed during the growing season when vegetation was actively assimilating carbon to sample a wide range of flux values (both $CO_2$ and energy components) for a robust comparison. For the ECOR systems in the SGP, this roughly corresponds to mid-March (start of growing season) through late June (senescence and dry-down of wheat crops). The wheat was in an early growth stage with little leaf area during first site visit at E41, matured and had much higher leaf area at the subsequent sites (E33 then E39), began forming grain heads while at E37, and was fully headed out and nearing senescence while at E14.

Overall, the energy fluxes showed good agreement between the two instrument systems during the campaign with an inter-site mean (range) Pearson's linear correlation coefficient ($\rho_P$) of 0.99 (0.99 – 1.00) and 0.92 (0.88 – 0.99), and bias (as quantified by the deviation of the orthogonal linear regression slope from unity, with > 0 indicating |ECORSF| > |reference|) of 1.0 % (– 1.6 – 3.3 %) and 8.8 % (– 3.2 – 27.3 %), for H and LE, respectively (Fig. 4; Table 4). Bias in $F_c$ were larger than for energy fluxes at 34.3 % (– 0.2 – 142.1 %), as was the scatter, with $\rho_P$ = 0.82 (0.61 – 0.98 %). As with the wind statistics (not shown), $u_*$ from the different systems compared well, but was generally smaller from the ECORSF: $\rho_P$ = 0.97 (0.95 – 0.99) and bias = – 10.0 % (– 15.7 – (-) 7.5 %). For sites visited later in the season (c.f., E41) there was, in general, better correlation in LE and $F_c$ (higher $\rho_P$), presumable due to the wheat crop at the sites being more mature and having a much higher leaf area than the early season growth at E41, which, in turn, was indicative of a higher growth rates and stronger signals in the fluxes involving water vapor and $CO_2$. E.g., when excluding analysis of E41, the $\rho_P$ increases to 0.93 and 0.88, and bias decreases in magnitude to 4.2 and 7.4 % for LE and $F_C$, respectively. However, no clear trend in bias was observed across the study period. It is noted that, unlike the above ECOR vs. EBBR comparison, to ensure more robust statistics, upwind vegetation type within the EC systems' footprints was not considered due to the short deployment durations. The magnitude of error resulting from the spatial displacement between the ECORSF and reference systems should only be critical in situations where there is significant heterogeneity of the fetch. This should not be the case for mature wheat crops, but may have been significant in the early season when spatially varying field conditions affected crop germination, growth, and evapotranspiration when leaf area is still small.

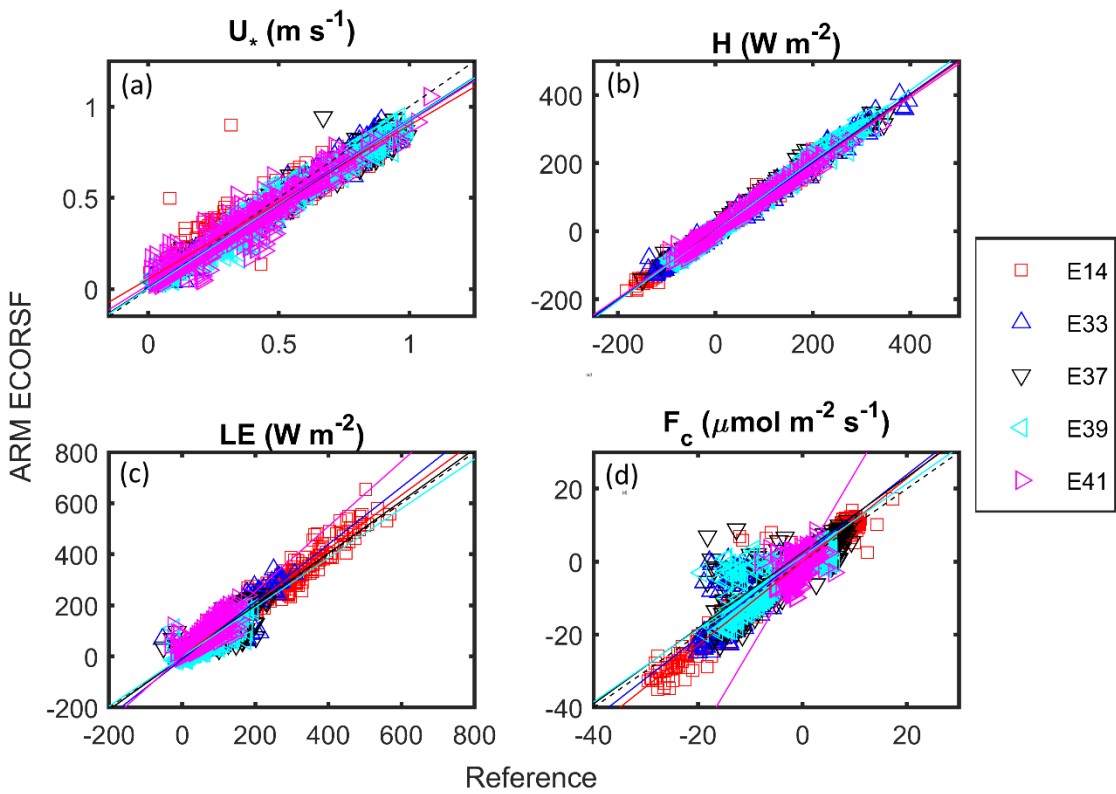

**Figure 4. Scatterplots of fluxes from the AmeriFlux[-like] portable eddy covariance reference system (abscissa) and**
**ARM ECORSF (ordinate) for 28 March – 21 May 2022. Colored markers and regression lines indicates the**
**individual deployments: E41 (Peckham, OK) 28 March – 6 April, E33 (Newkirk, OK) 7 April – 14 April, E39**
**(Morrison, OK) 15 April – 27 April, E37 (Waukomis, OK) 28 April – 9 May, E14 (Lamont, OK) 10 May – 21 May.**
**Also shown is a 1:1 line for reference (dashed black), and orthogonal linear regression lines (solid, colored by site).**

| Site | Dates | $\rho_P$ | | | | Bias (%) | | | |
|---|---|---|---|---|---|---|---|---|---|
| | | H | LE | $F_c$ | $u_*$ | H | LE | $F_c$ | $u_*$ |
| 41 | 28-March to 6-April | 1.00 | 0.88 | 0.61 | 0.96 | -1.6 | 27.3 | 142.1 | -10.4 |
| 33 | 7-April to 14-April | 1.00 | 0.92 | 0.88 | 0.99 | 0.6 | 12.6 | 11.7 | -7.6 |
| 39 | 15-April to 27-April | 1.00 | 0.91 | 0.85 | 0.98 | 3.3 | -3.2 | -0.2 | -7.5 |
| 37 | 28-April to 9-May | 0.99 | 0.89 | 0.81 | 0.98 | 3.0 | 2.2 | 3.7 | -8.6 |
| 14 | 10-May to 21-May | 0.99 | 0.99 | 0.98 | 0.95 | -0.1 | 5.3 | 14.3 | -15.7 |

**Table 4. Pearson's linear correlation coefficient ($\rho_P$) and bias (quantified using the deviation of the orthogonal linear regression slope from unity, with > 0 indicating |ECORSF| > |reference|) for the intercomparison between the co-located ECORSF (ordinate) and roaming reference (abscissa) EC systems at various SGP sites in 2022.**

## 5 Concluding remarks

The Atmosphere Radiation Measurement user facility's foundational objective is to improve the understanding of the influence of atmospheric radiation on atmospheric model performance via acquiring high fidelity, comprehensive in situ measurements of atmospheric state variables, from equator to poles (Stokes & Schwartz, 1994). Measurements of near surface turbulent fluxes quantify a key conduit between incoming and outgoing radiation from the Earth's surface, and its fate and role in atmospheric processes dictating weather and climate. Beyond the upward, atmospheric facing foci of ARM, fluxes mediate processes at the interface between the atmosphere, and the biosphere and land surface below. Since the early 1990's, ARM has measured these fluxes using two established methods, energy balance Bowen ratio and eddy covariance, at both long-term sites and shorter-term, mobile deployments. Herein, a summary of these measurements is provided, along with how these systems have evolved in time, documentation of general and specific aspects of the instrument systems and their data quality control, post-processing, and corrections, and general guidance of best use practices of the datasets. Additionally, results of three intercomparison validation exercises are presented to enhance confidence in the reliability of these datasets.

Consistent with previous literature (Barr et al., 1994; Billesbach et al., 2024; Tang et al., 2019a), LE estimated with the energy balance Bowen ratio method was larger than that measured with eddy covariance at SGP E39. This result is normally attributed to the EBBR system's forcing energy budget closure. This finding does not have any clear dependency on vegetation type (crop vs. grass). Smaller differences were observed between the two methods for H, and similar to LE, no vegetation type dependency was found. This should not be interpreted to mean that vegetation type does not influence the magnitude of the fluxes themselves (Williams & Torn, 2015), and when synthesizing these data for spatial averages, data users should be aware of the impacts both of instrument type and underlaying surface characteristics.

During testing of the new ECORSF prototype, it was deployed alongside the AmeriFlux site at the Fermilab Prairie (US-IB2). This flux system was built to the same specifications as the 2003/2004 ECOR systems, allowing an analogous pseudo-comparison between the two generations of ARM EC flux systems. Biases between the ECORSF and US-IB2 were generally within the estimated instrument uncertainty (Cook & Sullivan, 2025b) for H and LE, but a larger bias was observed for $F_c$, potentially due to the heterogeneity in

vegetation density and species within the prairie, even over the small separation ($\sim$ 5 m) between the two
systems. Consistent with the increased sensitivity of the newer IRGA models used in the ECORSF, daytime LE
and $F_c$ measured from this system were greater in magnitude.

After deploying the ECORSF across the SGP facilities, an additional portable flux system, akin to the
AmeriFlux PECS, was acquired and deployed for periods of approximately two weeks at each facility. As with
the comparison at US-IB2, H and LE measured by the ECORSF and the portable reference systems generally
agree within the expected measurements uncertainty, although slightly higher discrepancy was observed for $F_c$.
However, as expected, the instantaneous uncertainty in $F_c$ between the two systems generally decreased (higher
$\rho_P$) throughout the intercomparison as the wheat crops matured and increased in leaf area.

These intercomparison experiments are intended to aid in interpretation of fluxes measured between the two
methods used within ARM, and to provide confidence in the consistency and fidelity of fluxes measured by the
EC method. Herein we document the history of, best use recommendations for, and various matters of
consideration regarding ARM flux data. It is strongly encouraged that data users take this information into
account when analyzing and interpreting data from the instrument systems.

**6 Data availability**

The ARM data being presented herein is available, open, and free to use from the ARM data discovery
(https://adc.arm.gov/discovery/), under the Creative Commons Attribution 4.0 International License. Accessing
data from the ARM archives requires creating a free account with ARM. Per the registration page: "individual
demographic information will not be shared outside of ARM and DOE and the information in your ARM profile
is protected by the requirements established in the Federal Privacy Act of 1974. Aggregate anonymized
demographic information may be shared with confidential review committees who are charged to evaluate the
quality and efficacy of ARM. For example, summary statistics of all ARM users may be reviewed by the ARM
facility triennial review panel". While requested, questions regarding sex, race, ethnicity, and disabilities are all
either optional or have an option to not answer.

While not the primary dataset, nor focus the manuscript herein, the raw, high frequency data from the ECOR
sonics and IRGAs is freely available, as discussed in Sect. C4 "Raw, fast response sonic and IRGA data", from
ARM.gov or armarchive@arm.gov.

External data from AmeriFlux, used in the intercomparison in Section 4.2.2, is available from
https://ameriflux.lbl.gov/ (10.17190/AMF/1246066; Matamala, 2019) and data from the ARM ECORSF

prototype while deployed at Fermilab and the roving portable EC system while deployed at SGP are available

from https://zenodo.org/.

| | | doi | Reference |
|---|---|---|---|
| ARM datastream name | co2flx4m | https://doi.org/10.5439/1287574 | Koontz et al., 2015a |
| | co2flx25m | https://doi.org/10.5439/1287575 | Koontz et al., 2015b |
| | co2flx60m | https://doi.org/10.5439/1287576 | Koontz et al., 2015c |
| | 30co2flx4m | https://doi.org/10.5439/1989774 | Biraud & Chan, 2002a |
| | 30co2flx25m | https://doi.org/10.5439/1989776 | Biraud & Chan, 2002b |
| | 30co2flx60m | https://doi.org/10.5439/1992202 | Biraud & Chan, 2002c |
| | 30ebbr | https://doi.org/10.5439/1023895 | Sullivan et al., 1993 |
| | 30baebbr | https://doi.org/10.5439/1027268 | Gaustad and Xie 1993 |
| | 30ecor | https://doi.org/10.5439/1879993 | Sullivan et al., 1997 |
| | 30qcecor | https://doi.org/10.5439/1097546 | Gaustad 2023 |
| | ecorsf | https://doi.org/10.5439/1494128 | Sullivan et al., 2019a |
| | sebs | https://doi.org/10.5439/1984921 | Sullivan et al., 2010 |
| | amcmethane | https://doi.org/10.5439/1508268 | Billesbach 2012 |
| | co2flxsoil | https://doi.org/10.5439/1313010 | Koontz et al., 2015d |
| | co2flxrad4m | https://doi.org/10.5439/1313017 | Koontz et al., 2015e |
| | co2flxsoilaux | https://doi.org/10.5439/1313016 | Koontz et al., 2015f |
| | sgp30co2flx4mmetC1 | https://doi.org/10.5439/1989773 | Biraud & Chan, 2002d |
| Intercomparison datasets | | https://doi.org/10.5281/zenodo.14261417 | Sullivan et al., 2024 |

**Table 5. List of doi and references for dataset described herein.**

**Appendix A Acronyms and abbreviations**

ACT – Atmospheric data community toolkit

AERI – Atmospheric emitted radiance interferometer

AK – Alaska

AL – Alabama

AMC – AmeriFlux measurement component

AMCMETHANE - AmeriFlux and methane VAP

AMF[#] – ARM mobile facility [#]

ARM – Atmospheric Radiation Measurement [user facility]

BAEBBR – Bulk aerodynamic technique EBBR VAP

BNF – Bankhead National Forest

CEILPBLHT – PBL height derived from ceilometer

CF – Central facility

$CH_4$ – Methane

$CO_2$ – Carbon dioxide

CO2FLX – Carbon dioxide flux [measurement system]

$C_p$ – Specific heat of air

CRG – Coast-Urban-Rural Atmospheric Gradient Experiment (CoURAGE)

CSAPR – C-band scanning ARM precipitation radar

DL – Doppler lidar

DOE – [U.S.] Department of Energy

DQR – Data quality report

E39 – Extended facility 39

EBBR – Energy balance Bowen ratio [system]

EC – Eddy covariance

ECOR – Eddy correlation [flux measurement system]

ECORSF – ECOR with SmartFlux

EF – Extended facilities

ENA – Eastern North Atlantic

ESM – Earth system model

$F_c$ – Carbon dioxide flux

Fermilab – Fermi National Accelerator Laboratory

G – Ground heat flux

GCM – Global climate model

GNDRAD – Ground radiation system

GVR/GVRP – G-band vapor radiometer

H – Sensible heat flux

$H_2O$ – Water

IGBP – International Geosphere–Biosphere Programme

IRGA – Infrared gas analyzer

IRT – Infrared thermometer

KASACR – Ka-band scanning ARM cloud radars

KAZR – Ka-band ARM zenith radar

KS – Kansas

LE – Latent heat flux

LST – Local standard time

MCD12C1 - Terra and Aqua combined MODIS Land Cover Climate Modeling Grid (CMG) Version 6

MET – Surface meteorology system

MFRSR – Multifilter rotating shadowband radiometer

MODIS – Moderate Resolution Imaging Spectroradiometer

MWR – Microwave radiometer

NSA – North Slope of Alaska

OK – Oklahoma

OLI – Oliktok Point

PAR – Photosynthetically active radiation

PBL – Planetary boundary layer

PBLHTDL – PBL height derived from Doppler lidar

PBLHTMPL – PBL height derived from micropulse lidar

PBLHTSONDE – PBL height derived from radiosonde data

PECS – Portable eddy covariance system

QCECOR – Quality-controlled ECOR VAP

R – Net radiation

REBS – Radiation and Energy Balance Systems, Inc

RL – Raman lidar

RWP – Radar wind profiler

SEBS – Surface energy balance system

SGP – Southern Great Plains

SIRS – Solar infrared radiation station

SKYRAD – Sky radiation system

SONDE – balloon-borne sounding system

STAMP – Soil temperature and moisture profiles

SWATS – soil water and temperature system

T – Air temperature

T' – Instantaneous fluctuation of temperature about the mean

TDLS – Tunable diode laser spectrometry

u' – Instantaneous fluctuation of the horizontal wind speed component about the mean

$u_*$ – Friction Velocity

USA – United States of America

VAP – Value-added product

w' – Instantaneous fluctuation of the vertical wind speed component about the mean

WACR – W-band ARM cloud radar

X'$_c$ – Instantaneous fluctuation of mixing ratio of scalar "c" in air about the mean

X'$_v$ – Instantaneous fluctuation of mixing ratio of water vapor in air about the mean

XSACR – X-band scanning ARM cloud radar

XSAPR – X-band scanning ARM precipitation radar

β – Bowen ratio

$\overline{\Delta\rho_v}$ – Mean difference in water vapor densities between the upper and lower sensors

$\overline{\Delta T}$ – Mean temperature difference between upper and lower sensors

λ – Latent heat of vaporization of water (or the latent heat of sublimation for frozen conditions)

ρ – Density of air

ρ$_c$ –Density of scalar "c"

ρ$_P$ – Pearson's linear correlation coefficient

ρ$_v$ –Density of water vapor

τ – Momentum flux

$\sigma$ – Ratio of the densities of water vapor and dry air

μ – Ratio of molar masses of dry air and water vapor

**Appendix B Tables**

Table B1 Provides dates during which data is available from each respective instrument system and location.

Tables B2-B4 Provide a rough estimate of the vegetation type within the fetch footprint of the EBBRs (Table

B2) and ECORs at long term (Table B3) and mobile sites (Table B4). These data are compiled from a

combination of on-site observations during installation or site visits, site technician reports, and maps and

satellite-based imagery. Common crops across SGP include winter wheat, soybeans, alfalfa, sorghum, and corn,

but the specific crop planted varies season by season, and double-cropping is not uncommon.

Table B5 Provides available estimates of soil bulk density and texture at the ARM SGP sites.

| Site | Facility | ECOR | | ECORSF | | EBBR | | CO2FLX | |
|------|----------|------------|----------|------------|----------|------------|----------|------------|----------|
| | | Start date | End date | Start date | End date | Start date | End date | Start date | End date |
| anx | M1 | 5 Jan 2019 | 2 Jun 2020 | | | | | | |
| anx | S2 | 20 Jun 2019 | 2 Jun 2020 | | | | | | |
| asi | M1 | 27 Apr 2016 | 6 Nov 2017 | | | | | | |
| awr | M1 | 2 Apr 2016 | 1 Jan 2017 | | | | | | |
| awr | S1 | 12 Jul 2015 | 18 Jan 2016 | | | | | | |
| bnf | S10 | | | | | | | * | |
| bnf | S13 | | | 28 May 2025 | | | | | |
| bnf | S14 | | | 9 Apr 2025 | | | | | |
| bnf | S20 | | | 1 Oct 2024 | | | | | |
| bnf | S30 | | | 1 Oct 2024 | | | | | |
| bnf | S40 | | | 1 Oct 2024 | | | | | |
| crg | S2 | | | 1 Dec 2024 | | | | | |
| crg | S3 | | | 1 Dec 2024 | | | | | |
| crg | S5 | | | 1 Dec 2024 | | | | | |
| crg | S6 | | | 1 Dec 2024 | | | | | |
| cor | M1 | 23 Sep 2018 | 1 May 2019 | | | | | | |
| ena | C1 | 7 Mar 2014 | 10 Sep 2024 | 17 Sep 2024 | | | | | |
| epc | M1 | 10 Mar 2022 | 14 Feb 2024 | | | | | | |
| fkb | M1 | 14 Mar 2007 | 1 Jan 2008 | | | | | | |
| grw | M1 | 15 Apr 2009 | 11 Oct 2010 | | | | | | |
| guc | M1 | 15 Mar | 15 Jun | | | | | | |

| | | 2012 | 2023 | | | | | | |
|---|---|---|---|---|---|---|---|---|---|
| guc | S3 | 26 May 2021 | 16 Jun 2023 | | | | | | |
| hfe | M1 | 5 Jun 2008 | 28 Dec 2008 | | | | | | |
| hou | M1 | 16 Nov 2020 | 1 Oct 2022 | | | | | | |
| kcg | M1 | | | 21 Feb 2024 | | | | | |
| mao | M1 | 4 Mar 2014 | 1 Dec 2015 | | | | | | |
| nim | M1 | 26 Nov 2005 | 7 Jan 2007 | | | | | | |
| nsa | E10 | 16 Sep 2011 | 30 Sep 2024 | 10 Jan 2024 | | | | | |
| nsa | E11 | 26 Jun 2012 | 6 Dec 2016 | | | | | | |
| oli | M1 | 16 Jul 2014 | 15 Jun 2021 | | | | | | |
| pvc | M1 | 26 Jun 2012 | 29 Jun 2013 | | | | | | |
| pye | M1 | 2 Jan 2005 | 15 Sep 2005 | | | | | | |
| rld | M1 | 1 Oct 2005 | 28 Jan 2005 | | | | | | |
| sbs | M1 | 24 Sep 2012 | 28 Apr 2011 | | | | | | |
| sgp | C1 (4 m) | | | | | | | 18 Dec 2002 | |
| sgp | C1 (25 m) | | | | | | | 18 Dec 2002 | 20 Jul 2015** |
| sgp | C1 (60 m) | | | | | | | 1 Jan 2001 | 20 Jul 2015** |
| sgp | E1 | 3 Sep 2004 | 14 Oct 2009 | | | | | | |
| sgp | E10 | 10 Mar 2003 | 31 Aug 2011 | | | | | | |
| sgp | E11 | | | | | 4 Aug 2016 | 29 Sep 2023 | | |
| sgp | E12 | | | 10 Dec 2024 | | 29 Sep 1993 | 6 Dec 2024 | | |
| sgp | E13 | | | | | 20 Jul 1993 | 18 Dec 2023 | | |
| sgp | E14 | 9 Dec 2003 | 22 Oct 2019 | 31 Oct 2019 | | | | | |

| | | | | | | | | | |
|---|---|---|---|---|---|---|---|---|---|
| sgp | E15 | | | | | 11 Jul 1993 | 29 Sep 2023 | | |
| sgp | E16 | 25 Sep 2003 | 8 Jun 2011 | | | | | | |
| sgp | E18 | | | | | 10 Sep 1997 | 17 Nov 2009 | | |
| sgp | E19 | | | | | 30 May 1997 | 20 Sep 2011 | | |
| sgp | E2 | | | | | 22 May 1997 | 20 Oct 2009 | | |
| sgp | E20 | | | | | 6 Jul 1993 | 17 Nov 2011 | | |
| sgp | E21 | 2 Nov 2004 | 2 May 2019 | | | | | | |
| sgp | E22 | | | | | 4 Jul 1993 | 1 Dec 2009 | | |
| sgp | E24 | 18 Mar 2004 | 14 Nov 2009 | | | | | | |
| sgp | E25 | | | | | 10 Aug 1997 | 8 Apr 2002 | | |
| sgp | E26 | | | | | 5 Jul 1993 | 17 Dec 2009 | | |
| sgp | E27 | | | | | 7 May 2003 | 4 Dec 2009 | | |
| sgp | E3 | 3 Oct 2004 | 24 Oct 2009 | | | | | | |
| sgp | E31 | 15 Nov 2011 | 25 Oct 2019 | 25 Oct 2019 | 21 Sep 2021 | | | | |
| sgp | E32 | | | 11 Dec 2024 | | 28 Sep 2011 | 10 Dec 2024 | | |
| sgp | E33 | 15 Aug 2011 | 23 Oct 2019 | 23 Oct 2019 | | | | | |
| sgp | E34 | | | | | 2 Sep 2011 | 29 Sep 2023 | | |
| sgp | E35 | | | | | 5 Oct 2011 | 24 Sep 2023 | | |
| sgp | E36 | | | | | 28 Sep 2011 | 29 Sep 2023 | | |
| sgp | E37 | 29 Nov 2011 | 22 Oct 2019 | 22 Oct 2019 | | | | | |
| sgp | E38 | 19 Aug 2011 | 24 Oct 2019 | 24 Oct 2019 | 7 Jun 2021 | | | | |
| sgp | E39 | 10 Jun 2015 | 23 Oct 2019 | 23 Oct 2019 | | 30 Sep 2015 | 17 Dec 2023 | | |
| sgp | E4 | | | | | 13 Jul | 26 Sep | | |

| | | | | | | | | | |
|---|---|---|---|---|---|---|---|---|---|
| | | | | | | 1993 | 2011 | | |
| sgp | E40 | | | | | 15 Oct 2015 | 29 Sep 2023 | | |
| sgp | E41 | 26 Apr 2016 | 23 Oct 2019 | 23 Oct 2019 | 2 Aug 2023 | | | | |
| sgp | E5 | 9 Sep 2003 | 2 Nov 2009 | | | | | | |
| sgp | E6 | 15 Sep 2003 | 18 Oct 2011 | | | | | | |
| sgp | E7 | | | | | 4 Oct 1993 | 14 Nov 2011 | | |
| sgp | E8 | | | | | 12 Jul 1993 | 10 Nov 2009 | | |
| sgp | E9 | | | | | 11 Jul 1993 | 29 Sep 2023 | | |
| sgp | S4 | | | 4 Jul 2023 | 11 Sep 2023 | | | | |
| sgp | S6 | | | 4 May 2023 | 11 Sep 2023 | | | | |
| twp | E30 | 12 May 2013 | 10 Jan 2015 | | | | | | |
| twp | E31 | 4 Jan 2014 | 3 Jan 2015 | | | | | | |
| twp | E32 | 28 Mar 2014 | 0 Jan 2015 | | | | | | |

**Table B1. Dates of available turbulent flux measurements by site and instrument system type. *Site currently in installation phase. ** IRGA was removed from SGP CF at 25 and 60 m., but sonic remains active.**

| Site | Facility | Grass/Pasture | Crop |
|---|---|---|---|
| SGP | E2 | 71-137, 223-289 | |
| | E4 | 0-158, 202-360 | |
| | E7 | 0-244, 296-360 | |
| | E8 | 0-224, 314-360 | |
| | E9 | 0-360 | |
| | E11 | 0-360 | |
| | E12 | 0-360 | |
| | E13 | 0-52, 142-194, 328-360 | |
| | E15 | 133-360 | |
| | E18 | 138-325 | |
| | E19 | 0-133, 151-360 | |
| | E20 | 0-230, 310-360 | |
| | E22 | 0-49, 139-360 | |
| | E25 | 30-300 | |
| | E26 | 0-33, 243-360 | |
| | E27 | 20-156 | |
| | E32 | 0-360 | |
| | E34 | 0-360 | |
| | E35 | 0-360 | |
| | E36 | 0-360 | |
| | E39 | 0-80, 280-360 | 100-260 |
| | E40 | 0-360 | |

**Table B2. Direction of prevailing wind with sufficient fetch by predominant vegetation type for the EBBR systems at**
**SGP. Other wind directions are associated with fluxes that are affected by insufficient fetch and surfaces, buildings,**
**and vegetation that are not similar to the local field conditions.**

| Site | Facility | Grass/ Pasture | Crop | Other | Comments |
|------|----------|----------------|------|-------|----------|
| ENA | | 0-360 | | | Limited fetch in all directions |
| NSA | E10 | | | Tundra, 0-360 | 0-20 and 340-360 fetch is limited |
| | E11 | | | Saltwater sea, 0-100 and 350-360 | Other directions, beach gravel |
| SGP | A4 | | | | |
| | A6 | | | | |
| | E1 | | 0-53, 120-360 | | |
| | E3 | 0-48 | 132-260 | | |
| | E5 | | 80-260 | | |
| | E6 | 0-90 | 91-360 | | |
| | E10 | 0-360 | | | |
| | E14 | 352-85 | 129-265 | | |
| | E16 | 134-269, 334-360 | | | |
| | E21 | | | Forest, 0-360 | 0–30, the data may be suspect due to tower structure |
| | E24 | | 80-280 | | |
| | E31 | 30-80 | 100-200 | | |
| | E33 | 40-80 | 100-300 | | |
| | E37 | 280-310 | 135-260 | | |
| | E38 | | 150-260 | | |
| | E39 | 0-80, 280-360 | 100-260 | | |
| | E41 | 0-80, 280-360 | 100-260 | | |
| TWP | E30 | | | 0-100 and 145-360, Saltwater sea | |
| | E31 | 0-360 | | Also wetland, 0-360 | |
| | E32 | 0-360 | | | |

**Table B3. Direction of prevailing wind with sufficient fetch by predominant vegetation type for the ECOR systems at ARM long term sites. Wind directions not listed are associated with fluxes that are affected by insufficient fetch and surfaces, buildings, and vegetation that are not similar to the local field conditions.**

| Site | Facility | Grass/Pasture | Crop | Other | Comments |
|---|---|---|---|---|---|
| AMF | FKB, M1 | | | Unspecified | 40–159 and 176–209 fluxes are affected by insufficient fetch and surfaces, buildings, or vegetation that are not similar to the local field conditions |
| | HFE, M1 | 0-360 | | | |
| | NIM, M1 | | | Unspecified | 90–170 and 220–280 fluxes are affected by insufficient fetch and surfaces, buildings, or vegetation that are not similar to the local field conditions |
| | PYE, M1 | | | Unspecified | 66–92 fluxes are affected by insufficient fetch and surfaces, buildings, or vegetation that are not similar to the local field conditions |
| AMF1 | ANX, M1 | | | Ocean, 0-80, 180-225, and 315-360 | |
| | ANX, S2 | | | Unspecified | 270−360, fluxes are affected by insufficient fetch and surfaces, buildings, and vegetation that are not similar to the local field conditions |
| | ASI, M1 | | | Unspecified | |
| | COR, M1 | | | | 100-120 and 160-200, fluxes are affected by insufficient fetch and surfaces, buildings, and vegetation that are not similar to the local field conditions |
| | EPC, M1 | | | Ocean | 110-180, fluxes are affected by insufficient fetch and surfaces, buildings, and vegetation that are not similar to the local field conditions |
| | HOU, M1 | | | Unspecified | 30-150 and 300-330, fluxes are affected by insufficient fetch and surfaces, buildings, and vegetation that are not similar to the local field conditions |
| | MAO, M1 | 0-360 | | | |

| | | | | | |
|---|---|---|---|---|---|
| | GRW, M1 | 0-360 | | Also low shrub, 0-99 and 270-360 | |
| | PVC, M1 | 0-360 | | Shrubs, 0-360 | Some saltwater sea influence 0-100 |
| AMF2 | AWR, M1 | | | Snow and ice, 0-360 | |
| | AWR, WAIS, S1 | | | Tundra | Fluxes are affected by insufficient fetch and surfaces, buildings, and vegetation that are not similar to the local field conditions |
| | GUC, M1 | | | | Fluxes are affected by insufficient fetch and surfaces, buildings, and vegetation that are not similar to the local field conditions |
| | GUC, S3 | | | | 210-240, fluxes are affected by insufficient fetch and surfaces, buildings, and vegetation that are not similar to the local field conditions |
| | SBS, M1 | | | Snow, 0-360 | |
| | KCG, M1 | 0-360 | | | Fluxes are affected by insufficient fetch and surfaces, buildings, and vegetation that are not similar to the local field conditions |
| AMF3 | OLI, M1 | | | Tundra, 0-360 | Fluxes are affected by insufficient fetch and surfaces, buildings, and vegetation that are not similar to the local field conditions |

**Table B4. Direction of prevailing wind with sufficient fetch by predominant vegetation type for the ECOR systems at AMF deployments. Wind directions not listed are associated with fluxes that are affected by insufficient fetch and surfaces, buildings, and vegetation that are not similar to the local field conditions.**

| SGP extended facility | Bulk density (g cm$^{-3}$) | Soil texture |
| --- | --- | --- |
| E1 | 1.35 | Silt Loam |
| E2 | 1.08 | Silty Clay Loam |
| E3 | 1.29 | Silty Clay Loam |
| E4 | 1.59 | Fine Sandy Loam |
| E5 | 1.39 | Silt Loam |
| E6 | 1.32 | Silty Clay Loam |
| E7 | 1.34 | Silt Loam |
| E8 | 1.52 | Sandy Loam |
| E9 | 1.41 | Silt Loam |
| E10 | 1.34 | Clay Loam |
| E11 | 1.48 | Loam |
| E12 | 1.26 | Silt/Fine Sandy Loam |
| E13/14 | 1.4 | Silty Clay Loam |
| E15 | 1.55 | Loamy Fine Sand |
| E18 | 1.48 | Silt Loam |
| E19 | 1.4 | Silt |
| E20 | 1.39 | Silt/Fine Sandy Loam |
| E21 | 1.52 | Sandy Loam |
| E22 | 1.47 | Silt Loam |
| E25 | 1.43 | Loam |
| E26 | 1.75 | Fine Sandy Loam |
| E27 | 1.43 | Loam |
| E31 | 1.2 | Silt Loam |
| E32 | 1.31 | Silty Clay Loam |
| E33 | 1.3 | Silt Loam |
| E34 | 1.18 | Silty Clay Loam |
| E35 | 1.41 | Clay |
| E36 | 1.58 | Sandy Loam |
| E37 | 1.22 | Silt Loam |
| E38 | 1.39 | Silt Loam |
| E39 | 1.23 | Silt Loam |
| E40 | 1.37 | Silt Loam |
| E41 | 1.44 | Silt Loam |

**Table B5. Estimates of soil bulk density and texture at select ARM Southern Great Plains sites.**

**Appendix C Ancillary measurements**

Measurements of turbulent fluxes aid in quantifying the exchange of mass and energy between the Earth's surface and the overlaying atmosphere, and are thus strongly linked to processes occurring in the subsurface below and planetary boundary layer (PBL) above (Helbig et al., 2021). It follows that flux measurements can inform researchers studying processes within the subsurface and PBL, and vice versa, measurements of subsurface and PBL properties can inform researchers studying processes at the land(water)-biosphere-atmosphere interface. Unfortunately, comprehensive measurements across the Earth system continuum are expensive and resource demanding. However, large, centralized funding sources that pull on collective efforts across many participating institutions, such as the U.S. DOE ARM user facility, afford the opportunity to study processes across these scales. Below is a brief, and far from exhaustive, overview of some additional ARM measurements that may be of particular interest to the flux research community.

**C1 Surface Energy Balance System (SEBS)**

The EBBR systems, by definition, have perfect closure of the energy budget following Eq. 1. While Eq. 1 is incomplete and lacking storage, dissipative, and otherwise unaccounted for terms (e.g., below sensor canopy storage, metabolic processes, advective or dissipative fluxes, mesoscale circulations) (Butterworth et al., 2024), its application to measurements from eddy covariance systems can provide useful insight into uncertainty in the data (Franssen et al., 2010). Thus, even without an ideal method to correct observed energy balance deficiencies between turbulent heat fluxes (H and LE) and available energy (R and G) (Twine et al., 2000), measurements of R and G co-located with EC systems are desirable. Consequently, in 2010 ARM developed the Surface Energy Balance System (SEBS) and deployed these systems at all ECOR sites (Cook & Sullivan, 2025c; Sullivan et al., 2010) to measure the radiation and surface soil (ground) heat flux components of the energy budget.

Unlike the EBBR net radiometers, the SEBS radiometers partition measurements into incoming and outgoing, short- and long-wave radiation, separately. In the same configuration as the EBBR, the SEBS have soil heat flow plates at 5 cm depth, which are corrected for soil conductivity using soil moisture (measured in gravimetric units) measured at 2.5 cm depth, and estimate soil energy storage using soil temperature measured at 0-5 cm depth, along with the soil moisture measurement; the measured soil heat flux and soil storage are combined to compute the ground surface heat flux. Diverging from the EBBR set-up, only three sets of redundant sensors are installed within the radiometers' downward facing footprint in the SEBS.

As with the ECOR, over time a need to upgrade the SEBS was necessitated. The new SEBS systems, diverges only slightly from the original SEBS systems: the REBS, Inc. soil sensors were replaced by heat flux plates from Hukseflux, and the REBS, Inc. soil temperature and moisture probes were replaced by HydraProbe soil water sensors (combined temperature and moisture) from Stevens. The soil temperature is measured at 2.5 cm, c.f. 0-5 cm in the original SEBS, and due to the normally non-linear nature of soil temperature gradients, this may result in a biased soil temperature change, and thus soil heat storage, particularly when the surface is hot during the day or cool at night. Following the recommendation from the

heat flux plates' manufacturer, no correction for soil conductivity is applied to the soil heat flow measurements. This is expected to lead to an underestimation in the magnitude of the soil heat flux due to the 0 W m$^{-1}$ K$^{-1}$ thermal conductivity

reference used in their calibration (Hukseflux Thermal Sensors B.V., 2023). Contrary to the EBBR and original SEBS, the soil moisture measured by the HydraProbes is reported in volumetric units (where gravimetric soil moisture ≡ volumetric soil moisture / dry soil bulk density; see Table B8 for soil bulk density and texture at SGP). The new SEBS systems has been installed at the AMF3 deployment at BNF and AMF1 development in Baltimore, Maryland (CRG), while the old SEBS remain operational at the remaining ARM locations.

In addition to measurements of energy available for the turbulent fluxes, the SEBS also employs a Vaisala rain detector or "wetness" sensor. The wetness measurements provide a qualitative assessment of periods during which water (precipitation, dew, ice, etc.) may be accumulated on the IRGA optical path or to a lesser extent, the sonic transducers. The wetness sensor outputs an analog signal ranging from 1 – 3 V, corresponding to wet to dry conditions.

**C2 AmeriFlux Measurement Component (AMC)**

In developing sites with more comprehensive data records suitable for contribution to the AmeriFlux network (Billesbach & Sullivan, 2020a,b; Biraud et al., 2022, 2024; Sullivan et al., 2025a,b), ARM installed additional instrumentation at a few selected ECOR sites (NSA, SGP E39, and formerly at OLI): the AmeriFlux Measurement Component (AMC (Reichl et al., 2012)) system (Reichl & Biraud, 2016). These systems are similar to the SEBS, adding soil and radiometry measurements to aid in interpretation of ECOR datasets. Unlike the SEBS, which has soil sensors near the surface (< 5 cm depth), the AMC

deploys Campbell Scientific reflectometers (CS650L and CS655 depending on site) at two depths, approximately 10 and 30 cm, over 6 redundant, horizontally distributed locations to measure soil temperature and moisture (measured in volumetric units). As LE and F$_c$ are strongly dependent on vegetation activity, and although SEBS provides four-way radiometry measurements, the AMC also deploys photosynthetically active radiation (PAR) sensors to acquire measurements of up- and down-welling radiation available for vegetation use during photosynthesis.

**C3 Soil Water And Temperature System (SWATS) and Soil Temperature And Moisture Profiles (STAMP)**

Soil moisture is a critical variable in mediating land-atmosphere energy, water, and carbon exchange, influencing temperature and precipitation locally and downwind (Seneviratne et al., 2010). While all EBBR and post-2010 ECOR (from the co-located SEBS) have measurements of soil moisture near the surface at 2.5 cm depth, water available to plants for photosynthesis and transpiration typically resides deeper within the root zone. To fill this information gap, in 1996 ARM

began measuring soil properties across the SGP facilities using the Soil Water And Temperature System (SWATS; Kyrouac et al., 1996) (Cook, 2016b). These systems measured two redundant profiles of soil-water potential, soil temperature, and soil moisture at 5 – 8 depths from 5 – 175 cm, depending on site, using Campbell Scientific matric potential sensors (model 229L). In 2016, the SWATS began to be replaced by the Soil Temperature And Moisture Profile system (STAMP; Kyrouac et al., 2016) (Cook 2016b). As with the SWATS, the STAMP has redundant profiles (three for STAMP vs. two for SWATS)

but differs slightly with only 5 depths from 5 to 100 cm (or maximum depth available before bedrock) using Stevens HydraProbe soil sensors. Unlike the EBBR and original SEBS near-surface soil moisture that are measured in gravimetric units, soil moisture from the SWATS and STAMP are provided in volumetric units. Historically, the STAMP has only been deployed at locations across the SGP, but these systems will also be at the AMF3 BNF deployment.

**C4 Raw, fast response sonic and IRGA data**

A key feature of the ARM modus operandi is that despite the enormous economical and logistical cost of producing high quality atmospheric data in globally disperse, often harsh or remote locations, all data is openly available to anyone at no cost. Primary data from the EBBR, CO2FLX, and ECOR systems are 30-minute estimates of H and LE, and $CO_2$ (CO2FLX and ECOR only) and $CH_4$ (ECOR previously at NSA, AMF3's deployment in Oliktok Point, AK, and CO2FLX upcoming at BNF) fluxes, and various ancillary measurements such as surface radiation and ground heat flux, and atmospheric state 860 variables. All data is available in near real-time (generally within a few days) from the ARM data repository (https://adc.arm.gov/discovery/#/). While the primary objective of the CO2FLX and ECOR systems is to measure fluxes, the EC method requires the wind and scalar quantity measurements be made at high frequency (~10 Hz). The high frequency data itself, particularly wind (and thus turbulence), may be of scientific value to researchers for a variety of applications, but rapidly accumulates in terms of data volume, particularly over the vast site-years of ARMs operations, and is thus not 865 currently hosted publicly. However, all raw data is also freely available upon request to ARM via its website (ARM.gov) or email (armarchive@arm.gov), and raw data from the CO2FLX at SGP since 2015 is also available on the ARM data repository in the *.a0 datastreams (see Chan & Biraud, 2022 for list of datastream names).

**C5 Atmospheric measurements from the surface though the planetary boundary layer (PBL)**

ARM measures various additional measurements from the subsurface through the troposphere that can be useful in addition 870 to ARM turbulent flux measurements. While near surface meteorological variables, such as temperature, humidity, and wind speed and direction are available from the EBBR, ECOR, and CO2FLX, these are not primary variables from these systems and may not be the best suited for data analysis. E.g., sonic temperature is derived from the speed of sound (which is dependent on the virtual temperature, not ambient air temperature) as well as air density; EBBR temperature and humidity heights are not static, but change height every 15 min; the sensor configuration is not designed for measuring reference level 875 ambient conditions, with some sensors being located within instrument control boxes; etc. Further, other common meteorological variables, such as precipitation, are not available directly from the flux systems. Thus, data users can obtain high quality meteorological measurements from the ARM surface meteorology systems (MET (Kyrouac et al., 2021)). Moving away from the surface, profiles of meteorological state variables are routinely measured via radio sondes in the balloon-borne sounding system (SONDE (Keeler et al., 2022)). An exhaustive list of other measurements is beyond the 880 scope of this manuscript, and is variable depending on the specific site and is not temporally static throughout the history of ARM. In brief, flux scientists can also find estimates of PBL height derived from ceilometer (CEILPBLHT), micropulse

lidar (PBLHTMPL), Doppler lidar (PBLHTDL), or radiosonde data (PBLHTSONDE); fluxes of up- and down-welling radiation from the ground radiation (GNDRAD) and sky radiation (SKYRAD) systems, respectively, up- and down-welling, long- and short-wave radiation from the solar infrared radiation station (SIRS), and narrowband global and diffuse solar radiation from the multifilter rotating shadowband radiometer (MFRSR); surface and sky temperature from the infrared thermometer (IRT); vertical profiles of temperature and water vapor from the atmospheric emitted radiance interferometer (AERI) or Raman lidar (RL), water vapor from the G-band vapor radiometer (GVR/GVRP), liquid and vapor water from the microwave radiometer (MWR), wind and turbulence from the Doppler lidar (DL) or radar wind profiler (RWP); and various research radars such as the C-Band and X-Band scanning ARM precipitation radars (CSAPR and XSAPR), Ka-Band and X-Band scanning ARM cloud radars (KASACR and XSACR), Ka-band ARM zenith radar (KAZR), W-band ARM cloud radar (WACR); amongst a plethora of other datasets.

**Author contributions**

Conception, data analysis, and manuscript writing were conducted by RCS. SLS contributed figure design and manuscript writing. RCS, DPB, SB, SC, RH, EK, JK, SP, MP, AT, MT, and DRC contributed to instrument development, and data development and acquisition. RCS, DPB, SB, SC, EK, JK, SP, MP, SLS, AT, MT, and DRC contributed to manuscript revision.

**Competing interests**

The authors declare that they have no conflict of interest.

**Acknowledgements**

Observations from the Atmospheric Radiation Measurement (ARM) user facility are supported by the U.S. Department of Energy Office of Science user facility managed by the Biological and Environmental Research Program. Work at Argonne National Laboratory was supported by the U.S. Department of Energy, Office of Science, Office of Biological and Environmental Research, under contract DEAC0206CH11357. We acknowledge the AmeriFlux site, US-IB2, for their data (Matamala, 2019). In addition, funding for AmeriFlux data resources was provided by the U.S. Department of Energy's Office of Science. We also thank K. Gaustad, S. Tang, C. Tao, S. Xie, and Y. Zhang for their roles in processing the BAEBBR and QCECOR VAPs; K. Reichl, J. Howie, A. Moyes, and J. Curtis for their role in the AMC dataset; B. Ermold, A. Koontz, and Y. Shi for their role in datastream curation; J. Martin, J. Stow, B. Williams, and the other numerous site technicians for their continuing support; C. Martin, D. Busch, S. Sarvey, and G. Sawyer for logistical support; and S.

Abernethy, P. Lech, and D. Swank for their assistance with integrating the dataset described herein into the ARM infrastructure.

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
