# Peer review of "Over three decades, and counting, of near-surface turbulent flux measurements from the Atmospheric Radiation Measurement (ARM) user facility"

_Earth System Science Data, 2025_

## Author Comment (AC2)

**Response to reviewer's comments**

MS No.: essd-2025-168

Over three decades, and counting, of near-surface turbulent flux measurements from the

Atmospheric Radiation Measurement (ARM) user facility

by Sullivan et al.

**We thank Drs. Cox and Williams for their time and effort in reviewing our manuscript, and their comments which were helpful in improving the manuscript. Responses to specific comments are given below in bold.**

RC1

The manuscript "Over three decades, and counting, of near-surface turbulent flux measurements from the Atmospheric Radiation Measurement (ARM) user facility" by Sullivan et al. provides comprehensive documentation of surface-based turbulent flux observing at ARM. The manuscript reviews methods, history, configuration, validation, site characteristics, recommendations, available support tools, data access information, context with similar networks, etc. The manuscript provides invaluable documentation of a complex, global series of turbulent flux measurements that have been operated be ARM for more than 20 years.

I'm a user of ARM data myself, occasionally including the turbulent fluxes. While ARM data sets are known for exceptional documentation, the turbulent heat fluxes have been one of the more complex (in terms of varied application) and thus less tractable data sources provided by the organization. Therefore, this manuscript is a welcome addition to ESSD and will provide an excellent basis for researchers interested in the ARM turbulent flux products, making this one of the easiest manuscripts to judge in my career. The manuscript should be published promptly.

I have only one question (hopefully I didn't miss this). What happens to the raw, high-frequency (~10 Hz) component measurements (T,u,v,w,q)? Some researchers with specialized needs may be interested in the raw data to analyze spectral details or subsets over varying integration windows with various applications for corrections. Are the raw data available for these purposes? If they are archived, but unavailable, I recommend ARM consider releasing them with DOI (though please don't hold up publication of this manuscript to do so). If they are not archived, I recommend ARM consider doing so in the future.

ARM does save and archive the raw data from the sonic anemometers and gas analyzers. While the large file size and lack of conformity of the raw data with ARM data standards (ARM Standards Committee, 2020) preclude the data from being hosted on the ARM data discovery portal, as with the processed flux data, these data are freely available. This is discussed in 'Appendix C4 Raw, fast response sonic and IRGA data': "However, all raw data is also freely available upon request to ARM via its website (ARM.gov) or email (armarchive@arm.gov)". As these data are not the primary dataset being described in the manuscript, they were discussed in the section referred to above, not the main text or 'Section 6 Data availability'.

Upon revisiting this aspect of the manuscript, we agree with the sentiment that the availability of this data is not clearly presented in the initial manuscript submission. We have modified the manuscript to clarify that this data is archived and freely available. In section 3.2, paragraph 1:

"Eddy covariance has been widely adapted as the gold standard method for measuring atmospheric fluxes globally across numerous networks and individual PIs, as it is one of the only methods that measures H and LE both directly and independently (Baldocchi et al., 2001, 2024; Beringer et al., 2016; Chu et al., 2017; Yamamoto et al., 2005). Unlike the EBBR, in addition to H and LE, the fast response sonic anemometers and $H_2O/CO_2$ IRGAs (see Table 1 for make and model) used in the EC method afford the calculation of momentum and $CO_2$ flux across the ECOR sites. While not the primary dataset, nor focus the manuscript herein, these high frequency, raw data are archived by ARM and freely available (see Sect. C4 "Raw, fast response sonic and IRGA data"). Similarly, raw data from the CO2FLX is available on request, and raw data from SGP since 2015 is also available on the ARM data repository in the *.a0 datastreams (see Chan & Biraud, 2022 for list of datastream names)."
And in the Data availability section:

"While not the primary dataset, nor focus the manuscript herein, the raw, high frequency data from the ECOR sonics and IRGAs is freely available, as discussed in Sect. C4 "Raw, fast response sonic and IRGA data", from ARM.gov or armarchive@arm.gov."
Reference

**ARM Standards Committee (2020). ARM Data File Standards Version: 1.3, DOE/SC-ARM-15-004. Washington, D.C.: DOE ARM Climate Research Facility. Retrieved from https://www.arm.gov/publications/programdocs/doe-sc-arm-15-004.pdf on 4 June 2025.**

RC2

This manuscript provides a unique historical background and synthesis of methods and metadata that will be of great value to many researchers and practitioners using ARM flux data. The comparisons between instrument systems will be very useful for common problems that require combining datasets. The techniques and theoretical basis are well described. There is a good balance between providing specific details on ARM approaches and providing just enough background for context, without becoming repetitive with existing texts. The manuscript is well written, and I appreciate having had the opportunity to review it. I have only one major suggestion that is not absolutely critical but would be nice to have, and a few areas where I recommend minor edits.

Major comments:

My only suggestion that could be considered "major" is to include some discussion and a link to data on the canopy height, if it exists. This may not be a huge concern at most of the sites due to the short-statured vegetation, but it is certainly a factor at E21. I realize that one could back out canopy height using methods such as Chu et al. 2018, but having direct estimates would be beneficial. I also recall a discussion many years ago regarding vegetation growing too close to some of the sensors (related to the more general concern of sensors placed deep within the roughness sublayer). Any clarification on that problem (if it still exists) would be helpful. **We discuss this in section 3.5.1 "ARM does not routinely publish comprehensive, site specific or temporally variant vegetation characteristics", but then point the reader to potential external sources, "For a qualitative assessment of temporal phenology of the vegetation at a specific site, it is recommended that data users consult external datasets, such as vegetation indices from satellite-based sensors (e.g., from Landsat, Sentinel, MODIS, VIIRS) or other vegetation synthesis databases (e.g., United States Department of Agriculture's CropScape). Since 2002, visible and infrared imagery have been taken at the ARM CF crop field near the CO2FLX tower and are available through the PhenoCam**

**network sites "armoklahoma" (2002 – 2014) and "southerngreatplains" (2012 – present) (Seyednasrollah et al., 2018)"**

**Regarding the comment of vegetation too close to some sensors, we believe this is in reference to the EBBR system. As stated in the instrument handbook, "The heights of the [automated exchange mechanism] aspirators are different at different extended facilities and depend on maximum vegetation height." During bi-weekly site visits, technicians observed the current vegetation height relative to the sensor placement. If the sensors were found too near to the vegetation, they were adjusted accordingly. It is possible that there were periods when vegetation growth during the two weeks between site visits resulted in the issue described by the reviewer prior to the heights being adjusted during the subsequent visit. Unfortunately, these brief instances were not recorded with sufficient detail to flag specific times during which the situation occurred.**

Minor comments:

L 24: perhaps use a semicolon instead of the awkward double parentheses.

**We have modified the text to remove the double parenthesis:**

**"(primarily carbon dioxide, with methane fluxes measured at two locations to date)"**

L 60: These data have been used extensively to study a range …

Since it's not feasible to cite everyone, it may be best to use e.g. in front of the chosen citations so as not to imply that the publications are this few.

**Suggested change has been made.**

L 199: "Where w' is the instantaneous fluctuation of the vertical wind speed component about the mean"

Rather than having to state "about the mean" after each variable in this section, you could just define the perturbation as a departure from the mean, and thereafter use the word perturbation without repeating "about the mean".

**Suggested change has been made.**

L 265: v for versus – I believe the scientific convention is vs. - also to be consistent with line 455. Please check throughout as this occurs often.

**Suggested change has been made.**

L 441: site was excluded

**Suggested change has been made.**

L 472: postulated that differences between…

**Suggested change has been made.**

I think this section needs to be revised as it makes it sound like the differences between this study and Tang et al. 2019 may be larger than in reality. See further suggestions below.

There is little doubt that the differences at the CF site are attributable to the vegetation differences. The "postulated" wording makes this sound more dubious than may be intended, especially when paired with the text that follows. This is not to suggest that instrument differences aren't also contributing.

**We agree and see that our wording was ambiguous and may have implied a different meaning than intended. We have modified the text to be clearer:**

**"In their study, Tang et al. (2019a) attributed the differences between the EBBR and ECOR, in part, to differences in vegetation upwind of the two systems."**

L 480: "we conclude that the differences between LE measured by the two methods (EBBR and EC) are reflective of differences in the instrument systems themselves, not solely due to environmental factor" I think you mean "differences … at the E39 site … are due to differences in the instrument systems themselves, not solely…" Since Tang et al. also compared averages over all the EBBR sites against the ECOR averages, the current wording may be taken to imply that instrument differences dominate across the entire ARM SGP domain. This was a popular viewpoint until it was encouraged to check this assumption, leading to the paper by Tang et al. 2019.

**Suggested change has been made.**

L 475: "...but was no longer significant…"

You may want to tweak this wording, as I think they did make note of non-negligible differences. From Tang et al.: "Although surface difference is the major factor contributing to the flux differences, the above results also show that instrument difference is nonneglectable."

**We have reworded to better reflect the statement we are paraphrasing from Tang et al. who wrote, "when the fluxes are filtered by NE winds, that is, both ECOR and EBBR are downwind of grassland, fluxes agree quite well between ECOR and EBBR, and the differences are not significant; when the fluxes are filtered by SE winds (Figures 2e and 2f),**

that is, ECOR and EBBR are downwind of different surface types, the differences are significant in summer season for LH and in spring and early summer for SH".

**Our new text reflects that while there was no significant difference when over same vegetation, there was still a difference:**

**"Specifically, when the datasets were segregated by wind direction, the observed differences were significant when the upwind fetch differed between the two systems, but while nonnegligible difference were also observed when both systems had upwind fetch over the same vegetation (grass), they were no longer statistically significant."**

L 476 "However, no clear dependence of the agreement on vegetation type was observed at E39,"

I would remove the "However" because I don't find this statement to be contradictory to Tang et al. given what is discussed next regarding the much larger separation between the EBBR and ECOR at the CF and the fact that the EBBR is in a different field entirely. Actually, their Figure 3 is not too far off from the LE and H differences shown in this manuscript. **Suggested change has been made.**

As an aside, curiously, a much smaller difference was found between the EBBR and ECOR at Medford E32 in 2016 (see Bagley et al. 2017). Any thoughts on this? Either way, you may want to discuss this paper as it attempted something similar in comparing the two systems for the same vegetation type/fetch. Also, you may want to note that EBBR does not share exactly the same footprint as ECOR even if it is mounted at the same height and location, and this is a source of uncertainty in the comparisons.

**We have included discussion on this paper, and its findings, in our introduction to the EBBR/ECOR comparison in section 4.1:**

**"As discussed above, in general across the SGP, the EBBR were deployed within grassland and the ECOR were deployed on the northern edge of crop fields. Thus, while ARM has been making measurements to be representative of both grassland and crop fluxes, interpretation of these datasets to characterize the impact of vegetation type on near surface turbulent fluxes is confounded by the differing underlying instrument method used to acquire these datasets. This was briefly addressed in Bagley et al. (2017), where an eddy covariance system was collocated with the EBBR at E32 for 8 months in 2016, and the observed fluxes were compared. They found high instantaneous agreement ($R^2 = 0.79$ and**

**0.73 for H and LE, respectively) and a low bias (regression line within 3% of a 1:1 line) between the two instrument methods during midday and concluded that the instrument effects on fluxes are small and data from the two systems were suitable for use in a synthesis analysis."**

**We have also added a note regarding the difference in fetch between the two instruments even when collocated.**

**"Further, even when within the same vegetation field, the actual footprints being measured by each system are not exactly the same, with the ECOR fetch generally extending to a greater distance from its sensors than for the EBBR (Cook & Sullivan, 2025a,b)."**

L 601: "This finding does not have any clear dependency on vegetation type (crop v grass)."

Here again, it reads too generally in my opinion. Yes, I agree that the analysis at E39 demonstrates the EBBR/ECOR differences, and shows minimal influence of vegetation type. Beyond E39, however, these two instrument types were distributed unequally across vegetation types within the ARM domain(s). Table B2 makes this clear. When the vegetation type (LAI) and function (GPP) are considered, as in Williams and Torn (2015), it becomes clear that there is a definite influence of the underlying vegetation in general. Many researchers want to average the data to obtain a spatial mean, and they need to know how to do this in a way that both represents the mixture of land surface characteristics and deals with instrument differences. My concern with the current wording is that some researchers will choose one instrument type or the other, in order to avoid dealing with instrument bias, and then proceed to average over only ECOR or only EBBR sites, thinking that they are getting a good spatial representation. Unfortunately, this approach was used many times in the past. I recommend adding a statement to the effect that, when aggregating data spatially, one needs to factor in both the instrumental differences as well as differences in underlying land surface characteristics. I believe that this manuscript does a nice job of providing the information needed for individual PIs to combine datasets as they see fit for their purposes.

**We have added discussion of this to the concluding remarks:**

**"Smaller differences were observed between the two methods for H, and similar to LE, no vegetation type dependency was found. This should not be interpreted to mean that**

**vegetation type does not influence the magnitude of the fluxes themselves (Williams & Torn, 2015), and when synthesizing these data for spatial averages, data users should be aware of the impacts both of instrument type and underlaying surface characteristics."**